# Movement Ecology of Adult Western Corn Rootworm: Implications for Management

**DOI:** 10.3390/insects14120922

**Published:** 2023-12-03

**Authors:** Thomas W. Sappington, Joseph L. Spencer

**Affiliations:** 1Corn Insects and Crop Genetics Research Unit, United States Department of Agriculture, Agricultural Research Service, Ames, IA 50011, USA; 2Department of Plant Pathology, Entomology and Microbiology, Iowa State University, Ames, IA 50011, USA; 3Illinois Natural History Survey, Prairie Research Institute, University of Illinois, Champaign, IL 61820, USA

**Keywords:** *Diabrotica virgifera virgifera*, resistance, dispersal, partial migration, ranging, station keeping, flight, behavior, Reid’s paradox, Slatkin’s paradox

## Abstract

**Simple Summary:**

The western corn rootworm is a destructive and mobile insect pest of corn in North America and Europe. It is difficult to manage, in part because resistance has evolved to many forms of control. Understanding spatial patterns and distances of adult flight is critical to improving pest and resistance management strategies. However, a holistic understanding of adult rootworm movement has remained elusive because of conflicting observations of short- and long-distance lifetime dispersal, a type of dilemma in ecology called Reid’s paradox. Estimates of gene exchange between populations provide indirect estimates of dispersal distances, suggesting movement that is much farther than that measured using direct field observations, a similar type of dilemma called Slatkin’s paradox. Taken together, the evidence is clear that many individual rootworms do not travel very far in their lifetime, often laying eggs in the same field in which they emerged. However, a substantial number of others take long flights of many kilometers before leaving offspring. We conclude that western corn rootworm is a partially migratory species consisting of two distinct behavioral types, *residents* and *migrants*. This interpretation will be useful in improving models of rootworm population dynamics and devising better rootworm pest management methods.

**Abstract:**

Movement of adult western corn rootworm, *Diabrotica virgifera virgifera* LeConte, is of fundamental importance to this species’ population dynamics, ecology, evolution, and interactions with its environment, including cultivated cornfields. Realistic parameterization of dispersal components of models is needed to predict rates of range expansion, development, and spread of resistance to control measures and improve pest and resistance management strategies. However, a coherent understanding of western corn rootworm movement ecology has remained elusive because of conflicting evidence for both short- and long-distance lifetime dispersal, a type of dilemma observed in many species called Reid’s paradox. Attempts to resolve this paradox using population genetic strategies to estimate rates of gene flow over space likewise imply greater dispersal distances than direct observations of short-range movement suggest, a dilemma called Slatkin’s paradox. Based on the wide-array of available evidence, we present a conceptual model of adult western corn rootworm movement ecology under the premise it is a partially migratory species. We propose that rootworm populations consist of two behavioral phenotypes, *resident* and *migrant*. Both engage in local, appetitive flights, but only the migrant phenotype also makes non-appetitive migratory flights, resulting in observed patterns of bimodal dispersal distances and resolution of Reid’s and Slatkin’s paradoxes.

## 1. Introduction

The western corn rootworm, *Diabrotica virgifera virgifera* LeConte (Coleoptera: Chrysomelidae), is the most significant pest of corn in North America and Europe [1], responsible for over USD 2 billion in combined management costs and yield losses annually in the U.S. alone [2]. Injury is caused mainly by larvae feeding on roots, disrupting water and nutrient uptake, and structurally weakening the plant, which may lodge in storms. Western corn rootworm biology and ecology evolved to exploit the annual availability of its host plant corn (or maize), *Zea mays*. This species has only one generation per year, overwintering in the soil as a diapausing egg [3,4]. Although the larvae can develop to adulthood on the roots of several wild grasses [5,6,7,8], for all practical purposes, they survive in significant numbers only on corn roots [1,9,10]. Thus, rotation of a field from corn, where eggs are laid one year, to another crop such as soybean (*Glycine max*) the following year effectively purges that field of larvae [11]. Annual crop rotation was recognized as an effective management tool since the earliest days of the western corn rootworm’s emergence as a pest [12] and remains an important option today [13,14,15,16,17]. In non-rotated corn in North America, larval control with soil insecticides was the most common management tool in the last half of the 20th century [18], until the advent of transgenic Bt corn in 2003 [19]. Control of adults with foliar insecticides was a common tactic in parts of the Great Plains as well [18].

However, nearly every management tactic deployed against western corn rootworm has been compromised by evolution of resistance. For example, adult females in most populations oviposit predominantly in cornfields; however, important exceptions are rotation-resistant populations found in parts of Illinois and surrounding states, characterized by females with relaxed oviposition fidelity to cornfields [1,11,13,20,21]. A capacity for adult movement at multiple spatial and temporal scales has facilitated the rapid evolution and spread of resistant populations. Having a solid, comprehensive framework of rootworm dispersal is critical for efforts to understand and predict the ecological and demographic consequences of adult movement under different scenarios of biotic and abiotic conditions, landscape matrices, and human interventions. The goal of this paper is to review and synthesize the wide array of available evidence illuminating adult western corn rootworm dispersal activity as the basis for a proposed conceptual model of this species’ adult movement ecology.

### 1.1. The Paradox of Western Corn Rootworm Movement

Many, perhaps most of the puzzle pieces needed to reveal the full picture of western corn rootworm adult movement have been painstakingly gathered over decades of research by numerous scientists. But how to fit all those pieces together into a coherent whole has remained frustratingly elusive. The challenge lies in reconciling incongruities between direct observations of readily measured, short-distance movement with indirect but compelling evidence of long-distance displacement of rootworm adults. Such a dilemma is called *Reid’s paradox* [22]. Reid [23] pointed out that as the glaciers retreated at the end of the Pleistocene, trees expanded northward in the British Isles at a much faster rate than seemed possible based on observations of seed dispersal mechanisms. In addition to plant range expansions (e.g., [24]), Reid’s paradox has been encountered in other taxa and ecological contexts as well [25,26,27,28].

Short-distance lifetime dispersal by western corn rootworm adults is well supported by both direct and indirect evidence. Based on capture of adults self-marked with Bt-corn tissue in the gut, Spencer et al. [29] determined 85% of adults in corn moved 4.6–9.1 m/d on average. Larval population density builds over consecutive generations in fields of continuously planted corn [4,30,31,32,33,34,35,36]. For density to build in cornfields over time, a large proportion of the adults emerging in a field must also lay many of their eggs in the same field. Furthermore, resistance to Bt toxins such as Cry3Bb1, expressed in transgenic Bt corn specifically targeting corn rootworm larvae, appears to have emerged independently in multiple locations across the Corn Belt, creating resistance hotspots in the landscape [37,38,39,40]. Resistance to a single-toxin Bt-corn hybrid can evolve in as few as three generations under artificial selection in the laboratory or greenhouse [41,42], or naturally in continuous Bt cornfields [34,37,38,43,44]. Fast evolution of resistance in the field is facilitated by assortative mating brought about in part by incomplete dispersal of adults from the field before mating and oviposition [19,45,46]. Shrestha and Gassmann [47] found that a western corn rootworm population in a field with a long history of Cry3Bb1 resistance had higher survival than rootworms from a field with a more recent history of resistance. This finding is consistent with a build-up of resistance in a field over time, indicating emigration of resistant beetles from that field is not complete and immigration of susceptible beetles is not sufficient to counteract the increasing frequency of resistance alleles in situ.

At the same time, evidence for emigration of western corn rootworm adults out of their natal field is also robust. Except in regions where rotation resistance is common [1,13,48], virtually no western corn rootworm adults emerge from first-year cornfields. Thus, adults found in a first-year cornfield are immigrants from elsewhere. Colonization of and oviposition in first-year cornfields is evidenced by accumulation of adults during that first year and by larval injury and adult emergence in second-year cornfields [30,49,50,51,52]. Similarly, an aerial insecticide application to suppress ovipositing western corn rootworm adult populations can protect a cornfield from economic injury by larvae the following year [18,35,53,54,55,56]. Pruess et al. [53] concluded that fields in central Nebraska treated for ovipositing adults the previous year were repopulated the next year in part by immigrant adults. Although the origin of immigrants recolonizing a depopulated field is primarily from nearby fields [50,52,57], there is evidence for dispersal of western corn rootworms well beyond the immediate surroundings of the natal field. This evidence includes rates of range expansion and spread of adaptive traits [1,4], ascent of freshly mated females into the atmosphere [58], wash-ups of adults on the shores of Lake Michigan [59,60], tethered-flight experiments [61], and population genetic estimates of gene flow [62], all of which will be examined later in this paper.

Long-distance flight by at least some western corn rootworms is not in doubt. Thus, in one respect, the solution to Reid’s paradox is simplified—it seems obvious that dispersal in this species is bimodal, in the sense that some individuals remain throughout their lifetime to reproduce in or near the natal field, while others fly long distances and reproduce elsewhere. However, recognition of a bimodal pattern of flight distances raises important questions. What constitutes a long-distance flight (how “long” is “long”)? What proportion of a rootworm population engages in long-distance flight? What motivations trigger a long-distance flight—i.e., what determines whether an individual takes only short-distance flights during its lifetime versus taking one or more long-distance flights? Is the timing of a long flight related to age or reproductive development? How many bouts of long-distance flight can be expected of an individual? Is migratory behavior involved? What external factors impact propensity and capacity to engage in a long-distance flight? We have clues and even answers to many of these and other questions for western corn rootworm, but before reviewing them, it will be helpful to briefly present some relevant concepts and terms used to describe insect movement and how these apply in general to western corn rootworm.

### 1.2. Types of Movement—Scale and Motivation

In a landmark paper, Nathan et al. [63] presented a unifying conceptual framework termed “movement ecology” for studying and understanding organismal movement. They describe four mechanistic components that interact to generate an observed movement path of an individual over a defined timeframe (from a few seconds to a lifetime), namely (1) internal state—the motivation to move; (2) motion capacity—the ability and modality of movement; (3) navigation capacity—the ability to direct movement toward a goal or target; and (4) external factors, which include any biotic or abiotic environmental conditions that affect the movement path directly or indirectly through their influence on the other three components. For many applications of this information, such as predicting population dynamics in a pest management context, or modeling evolution of resistance to a control tactic, the goal is to understand population-level patterns of movement and their impacts on the phenomenon of interest. Such population-level phenomena are emergent properties of individual behaviors and individual responses to selection [46,64,65,66,67]. Understanding the effects of dispersal on ecological and evolutionary dynamics depends on a robust understanding of the underlying mechanisms of individual movement [63].

All four mechanistic components of western corn rootworm movement by flight vary depending on the spatial and temporal scales of focus, as do the manner and consequences of their interactions at a given time and place. For example, the internal motivation for flight activity by an individual changes with intrinsic variables like age and physiological status (nutrition, mating, past flight activity, and reproductive development), and with extrinsic variables (biotic and abiotic) such as population density, crop maturity, and ambient weather conditions [68]. Fixed interindividual differences in characteristics like sex, genotype, and morphology can affect internal motivation or capacity for flight. Targets of navigation differ depending on motivation, such as the search for resources. The convergence or divergence of individual movement paths [63] at the population level can even affect extrinsic factors, which in turn can influence future flight behavior of individuals through density-dependent mechanisms.

Motivations underlying insect flight can be categorized as either *appetitive* or *non-appetitive*. Most flight activity is appetitive in nature, meaning that the individual is foraging or searching for a resource such as a mate, food, oviposition site, or shelter. Appetitive flight behavior is triggered by immediate conditions experienced by the individual and is arrested upon encountering the resource being sought [64]. Appetitive flight can be subcategorized based on the resulting degree of net displacement relative to the individual’s *home range*, the area in which day-to-day maintenance and/or reproductive activities occur. *Station-keeping* behaviors involve localized movement within the home range, particularly foraging activities [64,69]. For western corn rootworm, station-keeping behavior often applies to movement within a single field and fields in the immediate vicinity, together constituting the individual’s home range. *Ranging* is appetitive flight behavior that results in a permanent displacement out of the previous home range [69,70,71] and can be thought of as extended foraging for whatever resource is motivating flight at the moment.

For example, an adult western corn rootworm leaving a senescing cornfield in appetitive flight to search for later-planted, more attractive corn [49,72,73] may disperse across the road in a station-keeping foraging flight or across a considerable distance in ranging flight before finding such a habitat. Whether the individual leaves its current field because it is pushed by deteriorating conditions [21] or pulled by detection of relatively more-stimulatory volatiles emanating from a less mature cornfield [74] is unresolved, but the nature of the flight is appetitive regardless. Even if displacement occurs over a longer distance, such a ranging flight ends as soon as the sought-after resource is encountered. The beetle then resumes normal day-to-day station-keeping behaviors in its new home range, which now includes the location where its resource-seeking ranging flight terminated.

At a behavioral level, station-keeping and ranging flights are fundamentally the same in the sense that they both reflect appetitive behavior. The categories are descriptive, in that they are distinguished by whether the distance traversed is great enough to preclude re-encounter with the field of origin during subsequent station-keeping flights; if so, the individual has left its home range, and if not, it remains within its home range. Thus, only the distances traversed distinguish these two categories, and even then, displacement distances via station-keeping and ranging behaviors may grade into one another [64]. Nevertheless, these categories are useful, because the distances traveled are determined by the insects’ responses to their proximate environment, which are important to understand for predicting movement patterns within and between fields in the local landscape. The consequences of the distances traveled are also of practical importance for population management, crop management, and insect resistance management (IRM). Even in seemingly homogenous agricultural landscapes, like those in the Corn Belt, ranging and some station-keeping movement likely result in displacement exposing individuals to fields—i.e., local environments—that vary in the particulars of seasonal and even day-to-day management. Such variables include which Bt/RNAi traits are expressed; crop phenology; local population density; the likelihood that a field may be sprayed or not at pollination; and weed, disease, and other pest pressures. The western corn rootworm “experience” can be quite different on the other side of the road or fence line.

In contrast, *migratory* flight behavior is non-appetitive. Migratory flight is not initiated as an immediate response to lack of a resource, and a migrating individual is not distracted by encounters with resources. It is characteristically persistent, straightened-out (i.e., non-meandering, non-searching) flight, which is terminated in a systematic way by environmental or internal cues unrelated to resource cues [64,66,75,76,77,78]. Migratory flight in insects that utilize winds at altitude for transport is characterized by three phases [79,80]: (1) *ascent* into the atmosphere above the flight boundary layer; (2) *transmigration* during horizontal displacement, characterized by maintenance of straight-line flight, altitude, and suppression of response to resource cues; and (3) *termination* of migratory flight accompanied by descent and landing. After termination of migratory flight, the insect finds itself in (or over, if the switch occurs while still airborne) a new environment, which may fortuitously be a habitat suitable for a new home range where it can begin station-keeping activities. However, if the habitat is unsuitable, the individual may, in some species, recommence with another bout of migratory flight, or it may begin appetitive ranging behavior in the local landscape in search of favorable habitat.

Migration is a phenomenon associated with a *migratory syndrome*, a suite of developmental, physiological, morphological, behavioral, and life-history traits that together direct and accomplish successful migration [64,77,81]. The migratory syndrome is underlain by genetically controlled mechanisms shaped by natural selection. Although migratory flight can be directional and long-distance, it does not have to be either. Western corn rootworm is not commonly thought of as a migratory insect because this species’ overwintering and breeding ranges are the same. This differs from more familiar insect migrants like fall armyworm (*Spodoptera frugiperda*) [82,83,84,85] and monarch butterfly (*Danaus plexippus*) [86,87], which engage in long-distance migration between geographically disjunct overwintering and summer breeding ranges. However, insect migration is not a phenomenon limited to directional movement between seasonal ranges [75,88,89], and there is considerable evidence for western corn rootworm migratory flight behavior, which we present later in this review.

With this background in mind, we can better describe the overall nature of the bimodal dispersal pattern observed in western corn rootworm populations (Figure 1). Some, but not all, western corn rootworm adults migrate, the defining characteristic of a *partially migratory species* [65,67,90,91]. Partial migration is the most common type among migratory animal taxa [65,90,92]. We infer two behavioral phenotypes of western corn rootworm adults, *resident* and *migrant*. During its lifetime, a *resident* engages in flight only in the context of station-keeping and ranging behaviors, that is, only in appetitive flight. Rootworm females that oviposit all their lifetime eggs in the natal field and its immediate vicinity (*natal home range*) are residents. Individuals that travel far enough via appetitive ranging behavior to establish a new home range outside their natal home range are also residents. The distinguishing characteristic of a western corn rootworm *migrant* is that during its lifetime, it engages in at least one non-appetitive flight consistent with this species’ migratory syndrome. Not all aspects of the rootworm migratory syndrome are understood, but it includes behaviors such as purposeful ascent into the atmosphere above the flight boundary layer (altitude at which wind speed exceeds unaided flight speed of the insect [93]), and developmental timing in which young females migrate after mating but before egg maturation. Both are attributes of the migratory syndromes of many other migratory insect species as well [66,69,94].

The mixture of migrants and residents in a western corn rootworm population may have arisen as a bet-hedging strategy. A female risks a catastrophic loss of long-term fitness when ovipositing all its eggs in a single field (“putting all its eggs in one basket”) destined for rotation to a non-host crop, whereas a female that oviposits in more than one field, only some of which may be destined for crop rotation, virtually ensures some of its offspring will emerge in a cornfield the next year and thus survive to reproduce. Bet hedging is an evolutionary strategy to spread risk in unpredictable environments by producing alternative phenotypes. The strategy involves accepting reduced mean within-generation fitness to increase geometric mean fitness over generations [95]. Producing some offspring that migrate from the natal field could be an adaptation in western corn rootworm to spread the risk of crop rotation of the natal field, or to a progenitorial host with a patchy distribution. It seems likely that selection will favor females of either phenotype that produce a mixture of resident and migrant offspring. The proportions among the offspring of an individual female most likely will depend on inherited environmental threshold responses ([91]; see Section 7.2). Among residents, a female that oviposits on the isolated progenitorial host plant on which it (and some number of siblings) developed may also doom the offspring to crowding on a locally limited food supply. Larval overcrowding has negative effects on mortality, development time, and size [96,97,98,99,100]. The female would do well to move before ovipositing, but such movement only needs to be local and at the within-field scale.

While bimodal dispersal is probably an adaptation, in part, to the unpredictability of crop rotation of the natal field, the rotation-resistant phenotype of western corn rootworm arose in eastern Illinois in response to a highly predictable crop rotation of nearly all fields of corn in the natal landscape. It is interesting that rotation-resistant western corn rootworm populations are also partially migratory, as evidenced by rates of range expansion correlated with prevailing wind direction, including stratified dispersal, and ascent of individuals into the atmosphere, as will be discussed (see especially Section 5.3 and Section 5.5).

Western corn rootworm movement ecology is complicated (Figure 1), but recognizing the distinction between residents and migrants provides a framework for assessing the wide array of observational and experimental data on rootworm dispersal and placing them in proper perspective. Characterizing the many aspects of the movement of both residents and migrants is a challenge. Adequate description and understanding of long-distance movement are particularly lacking [4]. Much of the difficulty in understanding long-distance movement arises from conflation of appetitive ranging flight with non-appetitive migratory flight, both of which displace an adult rootworm out of its natal home range (Figure 1) but which derive from fundamentally different motivations and behaviors. The consequences of bimodal intrapopulation variability in dispersal behavior of individual western corn rootworms both determine and introduce variation in patterns of net generational displacement and gene flow at the population level. Knowledge of such patterns can help us predict population changes in a particular field or set of adjacent fields, even if we do not fully understand the motivations and determinative dynamics at the level of individuals. In the context of short-term management of populations infesting cornfields, and of designing and implementing strategies to delay, contain, or otherwise mitigate resistance to control measures, we can benefit from recognizing movement patterns within fields, between fields in the local landscape, and across larger expanses of space.

### 1.3. Incorporating Short- and Long-Distance Dispersal in Population Models

Short-range daily movement distances are used to parameterize the dispersal component of most models exploring western corn rootworm resistance evolution [101,102,103,104,105,106] and resistance mitigation [107]. The values used are generally in the range of a few tens of meters per day, although local ranging among fields in the modeling landscape is often assumed (see [108] for a review). Short-distance movement is associated with station-keeping activities within fields and ranging activities between nearby fields. Heavy reliance on the short-range, diffusive movements of the resident portion of the population to parameterize flight distances in models of western corn rootworm population dynamics, ecology, and evolution means that the role of concomitant long-distance movement by migrants remains mostly unknown and unaccounted for. Modelers are not unaware of the occurrence and potential importance of long-distance flight by at least some adults, but such movement is ill-defined, and including it as a realistic parameter can be challenging. Nevertheless, a few attempts have been made, yielding insightful results.

Caprio et al. [109] incorporated long-distance dispersal indirectly in their model for evolution of methyl parathion resistance. Instead of assigning distances traveled during long-distance flights, they simply included the daily rate of adults of different developmental maturities dispersing out of the natal field to any one of the other 24 fields in the modeling landscape, based on the percentage of sustained fliers, presumably migrants, observed in tethered-flight experiments [61,110]. Most commonly, long-distance flight is included in models focused on geographic spread of a trait or range expansion of the species itself. Onstad et al. [111,112] modeled predictions of geographic spread of rotation resistance through the western corn rootworm metapopulation from its point-source origin in east-central Illinois. They incorporated a maximum distance of wind-aided flight of 33 km calculated from speed of long-duration (>30 min) flights on flight mills [61] and typical characteristics of summer wind and storm events in the central Corn Belt.

The invasion of Europe by western corn rootworm generated great interest in developing models to predict expansion into new areas [113,114], and to assess efficacy of potential containment strategies and tactics to slow the spread [115,116,117]. As in the case of IRM models, dispersal is a key parameter that helps drive these models’ output. But because the context of most European models relates to predicting or slowing range expansion, long-distance dispersal plays a prominent role in parameterization. With some exceptions (e.g., [117]), the approach taken to identify realistic values with which to parameterize long-distance dispersal components has been empirical in nature, using rates of range expansion already observed for this species during the ongoing invasion process. This approach is practical and does not require detailed understanding of underlying processes governing the expansion. Though conceptually straightforward, acquiring good estimates of long-distance dispersal based on observed rates of range expansion is not easy, as will become evident later. Nevertheless, the effort to obtain such estimates for both the North American and European expansions has helped clarify the movement ecology of western corn rootworm, including the interconnected contributions of residents and migrants to net rates of expansion.

Many details of western corn rootworm adult movement ecology, especially understanding motivations (and the mechanisms controlling motivations) for movement and dispersal over different scales and involving different behaviors, will take much focused research to sort out. At the same time, we already know a great deal about the spatial patterns of adult movement of this pest species (Figure 1). In this review, we focus on describing these movement patterns, and their implications for pest management, IRM, and predicted range expansions. Notably, the experimentation and observations that have helped elucidate patterns of movement and dispersal also give us much insight into the mechanisms and fundamental drivers underlying the movement ecology of individuals, as well as exposing critical gaps in our understanding that would benefit from future research.

## 2. Pre-Mating Movement

Western corn rootworm adult emergence begins in late June or early July. Adult males emerge ca. 5–6 d before females (protandry) [32,118,119] due to faster pre- and post-eclosion development rates [119,120]. Newly emerged males are not sexually mature, requiring 5–9 d to become responsive to female pheromone [121]. Delayed sexual maturity results in time for pre-mating male activity, including feeding and dispersal. In contrast, adult females are sexually mature upon emergence; 54% and 96% engaged in pheromone calling behavior during the 1st or 2nd day after emergence, respectively [122].

Post-emergence, pre-mating movement by western corn rootworm adults varies by sex. Newly emerged females rest on corn leaves an average of 0.78 m above the soil surface and move little before mating [3,123,124,125]. Most females release pheromone from a position near their natal plant [11,40]. Much male activity, including responses to pheromone, occurs below 2.0 m in the corn canopy [126,127]. Marquardt and Krupke [125] suggested that pre-mating males move more than females.

When directly observing individual behavior is not feasible, the movement patterns of many beetles can be tracked using a variety of markers. Spencer et al. [128] and Hughson [129] used two Cry proteins differentially expressed in rootworm Bt-corn or non-Bt-corn (“refuge”) hybrids as ingestible markers [29] acquired during normal feeding. Cry protein-containing tissue remained detectable in beetle gut contents for up to ca. 24 h post-ingestion [29,129]. The presence of hybrid-specific Cry proteins in the gut contents of beetles collected from Bt or refuge corn rows reveals intrafield movement between field areas. Marker and labeling studies of western corn rootworm recovered in mating pairs [128,129,130] revealed there can be substantial movement of both sexes before mating.

Western corn rootworm intrafield movement is greatest during the vegetative period of corn phenology [128,129]. Among individual beetles collected in cornfields, the proportion engaged in intrafield movement drops significantly from 0.22 during the vegetative period to 0.06 and 0.07 during the pollination and post-pollination periods [129]. The wide availability of nutritious pollen and silks are likely factors that reduce hunger-related intrafield movement during pollination and the early portion of the post-pollination period. Individual gut-content analyses of the partners from a mating pair reveal details of pre-mating movement. Like the overall field population, among males collected in mating pairs, the proportion that engaged in pre-mating intrafield movement (0.38) was greatest during the vegetative period of corn plant phenology [128]. Mating females also engage in intrafield movement; however, the proportion of moving females collected in mating pairs (0.09) was significantly less than that of their male partners (0.25) [128]. Hughson [129] reported similar proportions of moving males (0.246) and females (0.030) in mating pairs.

Among females, the likelihood of pre-mating intrafield movement may be influenced by female age and/or access to mating opportunities. While Spencer et al. [128] identified females that moved across Bt/refuge boundaries before mating, these intrafield moving individuals were almost exclusively (93.1%) mature and non-teneral (i.e., they emerged >24 h before mating). Scarcity of teneral individuals among moving females suggests that pre-mating intrafield movement is a phenomenon largely limited to females that remain unmated more than one day after emergence [128]. The implication is that in areas where mate-seeking males are not abundant, female mating may be delayed. Eventually, as unmated females move and enter areas of higher male abundance (e.g., refuges), they are intercepted by mate-seeking males and are mated as older, non-teneral adults.

Most western corn rootworm beetles in these studies did not cross between Bt and refuge areas of cornfields before finding a mate. In fact, mating pairs that included a beetle that engaged in intrafield movement (between refuge and Bt corn) were mostly detected within a few rows of the interface between those areas [128,129]. Despite only modest proportions of males and females engaging in pre-mating intrafield movement, the average movement rate of those individuals was 29.5 m/d (based on Figures 3 and 4 in [128]).

Taylor and Krupke [130] studied western corn rootworm dispersal and mating interactions in Bt and non-Bt refuge corn. They labeled non-Bt refuge corn plants with ^15^N, which was ingested by feeding larvae and was later detectable in the adults, enabling definitive identification of refuge beetles when present in mating pairs [130]. In their study, 41.5% of mating pairs included partners that originated from different areas of their study plots. However, like Spencer et al. [128] and Hughson [129], when mating pairs with beetles from two different areas of their fields were detected, they were found not far from boundaries between non-Bt refuge corn and Bt corn [130]. While some significant pre-mating movement occurred in males, and to a lesser extent in females, most mate-seeking western corn rootworm beetles did not move very far from their emergence location to find a mate [130].

## 3. Mating and Movement

Western corn rootworm mating behavior was described in detail by Lew and Ball [124]. Initial detection of sex pheromone prompts an excited response from males [131]. In a laboratory wind tunnel, rapid waving of antennae (antennation; also important in the mating sequence [124]) combined with orienting the body with respect to the pheromone source precede initiation of an upwind or lateral hovering flight toward the source [131]. A slow hovering flight while approaching a pheromone source is documented for the related species *Diabrotica balteata* [132] in which males responded to pheromone lures in soybean fields from as far as 49 m. A similar slow, hovering flight is commonly observed among western corn rootworm males approaching a calling female. Branson and Krysan [133] characterized western corn rootworm pheromone as “extremely efficient” but unnecessarily so for a species where the sexes are at high density in corn. They speculated such efficiency is evidence of rootworm evolution under conditions of much lower adult density. This suggestion is consistent with other life-history traits indicative of a species adapted to low-density adult populations, such as protandry, female calling from the natal host plant, and post-mating female migration. An environment with scattered perennial progenitorial hosts [133] would favor other characteristics present in western corn rootworm, including high adult mobility and sensitivity to host plant volatiles [74,134], oviposition near the host plant [135,136,137,138], and high sensitivity and responsiveness to root-produced volatiles among larvae with limited capacity for moving through soil [139,140].

Western corn rootworm mating may be observed throughout the period of adult emergence. During a 3–4 h mating [141], males transfer a large spermatophore to the female [142,143,144]. The western corn rootworm spermatophore may equal up to 9% of the male’s mass [143] and constitutes a significant male investment in the female. Some of the spermatophore components are incorporated into the eggs [144]. Most females are thought to mate only once [3,123], though some mate a second time in the laboratory [39]. Dissection of females from extensive field collections by Hughson [129] revealed multiple spermatophores in some and other evidence indicating at least 4–5% likely mate multiply. Males are also capable of multiple matings [143,145,146], though acquiring resources to provision large spermatophores may limit a male’s capacity in this regard, especially as cornfields mature. Kang and Krupke [146] found most males were capable of only 2–3 matings during an approximately 10–14-d reproductive period following an initial mating. Bermond et al. [147] found three-fold greater survival of females than males under starvation in the laboratory and hypothesized that female use of nutrients from the male spermatophore may explain this result. In contrast, Murphy and Krupke [144] found no difference in longevity of mated and unmated females under starvation conditions in the laboratory. The potential contribution of spermatophore components to the metabolic demands of female migratory flight has not been evaluated. Evidence of recent mating is strongly associated with female western corn rootworm engaged in migratory behavior [58,148,149]. However, any potential role for the mating act itself or spermatophore components in stimulating some females to engage in migratory flight is unknown.

## 4. Post-Mating Movement

After mating, male and female movement patterns may diverge. Males presumably continue mate-seeking behavior in areas where female emergence is ongoing. Among females, mating stimulates egg development [141,150,151], necessitating that they locate and feed on nutritious host plant tissues. Newly mated females require 6–21 d of feeding to mature their first batch of eggs [141,152]. It is during this post-mating, pre-ovipositional period that females of the migrant phenotype engage in non-appetitive migratory flight (as will be described in detail in Section 5). Resident beetles of both sexes engage in appetitive station-keeping and ranging flight behavior after mating, which may keep an individual in its natal field or may lead to displacement over a relatively short distance within the local landscape.

Areas in Bt cornfields with the greatest adult abundance, along with significant mating activity, frequently coincide with nearby non-Bt refuge rows that are the areas of greatest adult emergence [32,129]. Adult intrafield movement between blocks of non-Bt refuge and Bt corn becomes less likely with advancing corn phenology [129]. Among free-moving adults in Bt cornfields with structured refuges, the probability of intrafield movement measured during the vegetative period (0.27) dropped significantly during pollination (0.01–0.07) and post-pollination (0.01–0.07) [129]. Among the minority of beetles that moved each day between refuge and Bt corn areas, intrafield (refuge to Bt corn) movement rates (males: 28.83 ± 3.62 m/d (mean ± SEM); females: 23.19 ± 0.19 m/d) were similar to pre-mating rates reported by Spencer et al. [128], though they were not significantly different between the sexes [129]. This contrasts with Ludwig and Hill [153], who suggested males are more mobile within fields than females. The declining probability of intrafield movement after the vegetative period of corn phenology may be attributed to the abundance of corn silks and pollen in the current field, to which western corn rootworm respond strongly [74] and which can concentrate populations [154].

Quantifying interfield movement is complicated by the scale of distances over which movement must be tracked and the difficulty of identifying specific beetles among large populations [148]. Mark-recapture experiments provide direct evidence of the net displacement and origin of individual immigrants. However, decreasing density of marked individuals with increasing distance from the source location usually limits efficacy of such a strategy to the local landscape. Various marking methods have been used to identify western corn rootworm movers, including fluorescent powder [148,155,156], ingested Bt protein [29], and N-isotope signatures [157]. For example, Toepfer et al. [158] mass-marked adults in cages with fluorescent powder and documented recapture of 0.03% in small corn plots 300 m from their release points in steppe habitat in Hungary. Using ingested Bt tissue as a marker, Hughson [129] examined the relationship of corn phenology in the local landscape to interfield movement. Among males, the proportion engaging in local interfield movement (measured between adjacent cornfields) was significantly higher during the vegetative period (0.09) than during pollination (0.02) or post-pollination (0.03) periods; female interfield movement was not measured. Failure to detect increasing interfield ranging movement as post-pollination cornfields matured may be attributed to near-identical phenology among the hybrids in all fields where movement was monitored [129].

Phenology has a role in various aspects of interfield western corn rootworm beetle movement. Interfield movement toward a less mature cornfield can occur when there is variation in crop phenology among local fields. At field interfaces, western corn rootworms express a short-range flight orientation preference for flowering corn vs. pre- or post-flowering stages and for corn vs. other crops (i.e., soybean, wheat, or sweet clover) regardless of corn developmental stage [134]. By extension, this tendency to orient toward corn should help beetles that have moved out of cornfields to relocate them. Campbell and Meinke [9] reported that western corn rootworm beetles moved from cornfields into adjacent non-corn habitats (native upland and lowland prairie) when corn had pollinated. McKone et al. [159] reported similar western corn rootworm usage of tallgrass prairie. Contrary to most studies suggesting that females predominate among western corn rootworms leaving corn, Campbell and Meinke [9] join Moeser and Vidal [160] to suggest there is significant movement out of corn by males. Western corn rootworm attraction to relatively less mature corn plants [74] is likely a factor in many instances of local interfield movement. This phenomenon is the basis for using late-planted corn as a “trap crop” to attract and concentrate egg-laying beetles into specific cornfields, creating infestations for study purposes or as a tactic to diminish infestations in the fields from which beetles were attracted [72,161]. A response among western corn rootworm beetles to a specific stage of corn phenology, R2 (post-silking blister stage [162]), was documented by Pierce and Gray [21]. They showed that increased interfield movement by rotation-resistant western corn rootworm adults out of corn into soybean or by rotation-susceptible adults into late-planted corn was associated with onset of the R2 stage [21]. Females (of both rotation-resistant and -susceptible populations) predominated significantly among the interfield movers from the R2 stage onward [163]. Notably, while differences in phenology of adjacent cornfields made the late corn a competitive sink for egg laying with soybean among the rotation-resistant population, extreme phenological differences did not lead to egg laying outside of corn where rotation resistance was absent [21]. While western corn rootworm responsiveness to corn phenology facilitates highly adaptive beetle movement, ovipositional fidelity to corn in the face of an extreme phenological difference suggests that its relaxation [164] is unlikely to account for the origins of rotation resistance.

Indirect evidence of interfield movement is frequently inferred from the proportions of western corn rootworm males and females detected outside of their natal fields. Female-biased sex ratios among beetles collected in rotated corn (and other rotated crops, especially where crop-rotation-resistant western corn rootworm populations are present [1]) testify to the different probabilities of male and female interfield dispersal from their natal fields [21,30,49,158,163,165,166]. Furthermore, these probabilities vary with flight altitude. While females outnumber males in most collections of movers, males are still present among the beetles flying between fields; nevertheless, females predominate at higher altitudes to a great extent [20,58,59,126,148,149,167]. Interfield movement (including initiation of migratory behavior and high-elevation flight) is strongly periodic and gated by permissive environmental conditions typical of mid-morning and late afternoon/early evening [58,167,168].

Use of indirect evidence to quantify interfield movement can provide valuable insights into western corn rootworm movement ecology, but interpretation of results can be more fraught with difficulty than often realized. Inferences about interfield movement drawn from sampling methods such as trapping or visual counts of adults often rely on the assumption that shifting patterns of spatial abundance emerge from movement patterns of individuals engaging in local appetitive ranging or station-keeping behavior. However, the partial migratory nature of western corn rootworm means that some portion of immigrants to a particular field are of the migrant phenotype that originated from somewhere beyond the local landscape. Likewise, of those individuals that emigrate from a field, not all will disperse to nearby fields—some will exit from the local landscape entirely via non-appetitive migratory flight. Levay et al. [52] concluded from trapping studies of semi-isolated pairs of fields separated by distances of 1–1400 m (one field of first-year corn and one field of continuous-planted corn) that ~38% of the adults that emerged in the continuous-planted field emigrated to colonize the paired first-year cornfield as immigrants. The experimental design and interpretation of data relied on important assumptions: (1) Emigrants from the continuous cornfield all colonized the first-year cornfield. This implies movement trajectories of emigrants all in the same direction, which in turn implies attraction from a distance during appetitive ranging flight. (2) Immigrants to the first-year field all originated from the (paired) nearest continuous cornfield and not from outside the 3 km semi-isolation zone. These are reasonable assumptions if all emigration and immigration resulted from local, appetitive station-keeping or ranging flight. However, they are potentially problematic under the proposition that emigrants and immigrants are a variable mixture of residents and migrants engaged in appetitive and non-appetitive flights of differing distances, directions, and motivations. This was a demanding and exceptionally well-conducted study, which illustrates the point that wringing insights about western corn rootworm movement ecology from experimental and observational data is especially challenging given this species’ partially migratory nature.

Among the examples of interfield western corn rootworm movement, there are significant differences between populations. In the eastern Corn Belt, movement between corn and soybean fields by crop-rotation-resistant females happens in a very different context from the interfield movements of females from cornfield to cornfield. Interfield movement by rotation-resistant females that lay some of their eggs in soybean fields is the behavior that allows them to circumvent annual crop rotation. However, distinguishing between normal station-keeping or ranging flight that moves a beetle into a new cornfield and the interfield flight of rotation-resistant western corn rootworm beetles leading to egg laying in soybean is challenging and context-dependent.

Rotation resistance in this species is almost certainly a genetically determined trait, as evidenced by its geographic spread from a point source in Ford County, Illinois [1,13], and it is clear that loss of female fidelity to oviposition in cornfields is the basis of rotation resistance [11,20,21,169,170]. Although the mechanism underlying the loss of fidelity is still undetermined, it is likely related to differences in flight behavior. It seems plausible, for example, that the extremely high selection pressure imposed by high-frequency crop rotation across a vast landscape could act on heritable variation to generate the observed differences in flight propensity between rotation-resistant and susceptible populations [171], but this remains a hypothesis to be tested. Some oviposition in non-corn crops by both rotation-resistant and susceptible (wild-type) beetles is undoubtedly the result of station-keeping behavior along the edge of a cornfield. While investigating invasive western corn rootworm in Croatia, Barčić et al. [172] showed that suspicious larval damage in rotated corn could be explained by an interfield movement edge effect extending ca. 20 m into the previous wheat or soybean field from adjacent continuous corn. Damage in the rotated crop was a consequence of adults laying eggs along the edge of the rotated crop. Because the crop fields were small (50 m wide), oviposition around the perimeter generated a pattern of damage that could be incorrectly construed as the presence of a rotation-resistant population. There is evidence for movement and edge effects on a similar scale in other studies. Well before rotation resistance became evident, Shaw et al. [136] reported larval damage in rotated corn (after soybean) at locations within 12 rows (9.1 m) of the edge of the previously adjacent cornfield. Spencer et al. [29] reported that 85–93% of crop-rotation-resistant western corn rootworm beetle movement within cornfields and from cornfields into soybean fields occurred at rates of 4.6–9.1 m/d. In another study of rotation-resistant western corn rootworms, Schroeder et al. [173] suggested that when crop fields (corn and three rotated crops) were small (i.e., 0.06 ha or ca. 24.6 m × 24.6 m) and in close proximity, it becomes a trivial task for beetles to move short distances to enter and lay eggs in an adjacent cornfield plot. Thus, in spite of equally high western corn rootworm female abundance measured in plots of corn, soybean, and other crops, egg laying and subsequent root injury in the rotated cornfields (mostly from soybean) were significantly reduced compared to that in the corn plots. Rondon and Gray [174] also evaluated adult western corn rootworm abundance in corn and a variety of rotated crops using small (0.1 ha) fields in a 5 × 5 Latin square. While they found significantly more females in soybean, there were equal numbers of eggs laid in all crops. They interpreted these results as evidence of non-preference in egg laying.

Something beyond station-keeping edge effects must be responsible for oviposition throughout large commercial soybean fields, observed as widespread damage to first-year cornfields. Compelling evidence for robust attraction to soybean tissues is lacking. The initial flight from a cornfield into an adjacent soybean field may result from a simple expression of enhanced propensity for flight. Field and laboratory behavioral assays showed that western corn rootworm adults from rotation-resistant populations are more active and ready to initiate flight than those from wild-type populations under the same rearing and environmental conditions [171]. In addition, nutritional stress caused by feeding on soybean tissue increases activity and probability of female flight compared to females in adjacent cornfields [175,176]. Though beetles from crop-rotation-resistant populations have a variety of adaptations to blunt the negative nutritional consequences of soybean herbivory that are lacking in rotation-susceptible populations [177,178,179], consumption of poor food (soybean foliage) stimulates behavior leading to interfield flight. Once females are active in soybean, consequences of soybean herbivory increase the probability of egg laying and flight. Thus, movement may play a role in egg laying if a soybean field is sufficiently large that some females remain long enough for the negative consequences of soybean herbivory to stimulate oviposition [21,175]. Soybean herbivory-stimulated flight out of soybean fields may be the mechanism that eventually returns a female to a nearby cornfield where it can feed on host tissues, allowing maturation of eggs that may be laid in soybean on a subsequent return to the field [176]. While rotation-resistant western corn rootworm herbivory in soybean can sometimes be dramatic [13], the high soybean tolerance for defoliation [180] suggests rootworm activity is unlikely to affect soybean yield, without contributions by other resident soybean herbivores.

Unlike most interfield ranging flights by western corn rootworm, the conditions responsible for initiating interfield flight by crop-rotation-resistant females differ depending on whether they are moving from corn to soybean or *vice versa*. Interfield movement from soybean to corn may be interpreted as an expression of appetitive ranging behavior in response to urgent nutritional stress. In contrast, most western corn rootworm females collected flying into a soybean field were not capable of ovipositing immediately, and only 20% carried resources sufficient to eventually lay eggs [181]. Combined with the poor quality of soybean tissues as food for western corn rootworm, a simple appetitive argument for this type of interfield movement seems problematic. Perhaps the initial flight out of corn is a manifestation of “typical” station-keeping behavior in a more mobile western corn rootworm population, resulting in inadvertent arrestment in a soybean field where opportunistic herbivory leads to nutritional stress. Alternatively, increased flight activity of rotation-resistant beetles [171] may also represent non-appetitive behavior, where non-directional short-distance displacement is itself the goal. Such behavior would not be conventionally migratory because a migration syndrome promoting long-distance flight is not involved, but it also would not be conventional appetitive flight behavior. Though speculative, a similar type of short-distance non-appetitive flight behavior has been hypothesized in reproductive (non-migratory) generations of the monarch butterfly [182].

## 5. Long-Distance, Migratory Movement

### 5.1. Laboratory Tethered Flight Behavior

Coats et al. [61] conducted tethered-flight experiments with mated female western corn rootworm of different ages on rotary flight mills and observed a clear differentiation between beetles engaging in “sustained” flights of 42–230 min duration and “trivial” flights of 1–17 min. Only 15% (28) of the 183 tested females engaged in sustained flight during the 24-h test period, and these averaged 4.5 sustained flights, each of about 72 min on average. That percentage includes all ages tested (2–15 d post-eclosion), even though females engaged in no sustained flights after 9 d of age. When broken down, 21% of females aged 2–9 d made a sustained flight, and 31% of those tested at the age of peak sustained flight activity (5–6 d) engaged in such flights (Table 1). All beetles engaged in numerous trivial flights across all ages tested, averaging about 3.9 min each. Based on their results, Coats et al. [61] argued that sustained flights represent migratory, non-appetitive flights by migratory beetles, and conversely that trivial flights represent appetitive flights. There are several lines of evidence that can point to an individual insect’s flight behavior being migratory, and Coats et al. [61] provided supporting observations that make a convincing case in this regard for western corn rootworm. Much additional evidence for western corn rootworm as a migratory species has accumulated in the subsequent 35+ years. Using Coats et al. [61] as a jumping-off point, those data merit a detailed summary.

Coats et al. [61] designated a fight duration threshold of 30 min to delimit trivial and migratory flights of western corn rootworm, based on reported flight duration distributions in tethered-flight experiments with the large milkweed bug, *Oncopeltus fasciatus* (Hemiptera: Lygaeidae) [186], and the convergent lady beetle, *Hippodamia convergens* (Coleoptera: Coccinellidae) [187]. However, a flight duration threshold to distinguish migratory flights in the laboratory depends on the species and can be difficult to resolve. It is best determined based on observations of flight duration distributions in the test population, along with associations of categories of flight duration with behavioral, age, physiological, and developmental characters. There was a clear, broad break in flight durations (no flights of 18–41 min) among the female rootworms tested by Coats et al. [61], making the 30-min threshold applicable to their dataset. Coats et al. [151] used the same threshold to distinguish short and long flights in a follow-up study demonstrating the role of juvenile hormone in regulating sustained flight activity. Naranjo [110,183] measured flight duration of western corn rootworm on actographs, a device that allows a tethered beetle to fly upward in a vertical plane and return to the platform to end a flight by landing. Although flight distance cannot be measured on an actograph, flight duration and timing of flights can. He also found a distinct bimodal distribution of flight durations by mated females, with no flights occurring between 18 and 29 min [110]. He therefore adjusted the flight duration threshold distinguishing migratory flight to 20 min, which also was consistent with data reported by Coats et al. [61].

Stebbing et al. [185] used the same actographs to compare flight performance of methyl parathion-resistant and normal susceptible western corn rootworms. For the susceptible beetles, the authors observed a continuous distribution of flight durations instead of a distribution with a distinctive gap between trivial and sustained flights. They retained the 20-min threshold for sustained flight used by Naranjo [110,183] for purposes of group comparisons. The beetles were tested in two broad age categories of 3–10 d and 11–20 d, which did not differ significantly in duration of sustained flights, and the percentage of mated females making a sustained flight was reported only for both age groups combined (Table 1). Interestingly, although no females >9 d old made a sustained flight in the studies by Coats et al. [61,151], some did in the 11–20-d-old category of Stebbing et al. [185], as well as in the 10–17-d and 23–30-d age groups tested by Naranjo [110], and the 20–25-d age group in the untreated controls of Naranjo [183] (Table 1). Yu et al. [100] also did not observe a distinct bimodal distribution in trivial versus sustained flight durations when they compared flight behavior of mated, 6-d-old western corn rootworm females reared at different larval densities. Frequency of longest-flight duration had reached a low level by 10 min, so that was declared the duration threshold of sustained flight for treatment comparison purposes. However, data in the supplemental material indicate 23% of females engaged in a sustained flight ≥ 20 min. The lack of an obvious break in duration distributions in the studies by Stebbing et al. [185] and Yu et al. [100] suggests there may not be a strict duration threshold defining migratory flight under all conditions. Nevertheless, in these six studies and an unpublished dissertation [184], the percentage of mated females that engaged in sustained flights of at least 20 min were fairly similar and highest among those in age bins under 10 d, ranging from about 21% to 54% (Table 1). Also, the frequency distribution of flight duration in a test population was always positively skewed and leptokurtic, with many short bouts of flight, tapering off rapidly to a fat tail of progressively rarer, longer flights. In a laboratory study, Li et al. [188] found that 5-d-old adults had greater propensity to takeoff and a shorter time to takeoff than 30-d-old adults in free flight after release at the base of a vertical stick. While this is consistent with age differences in sustained tethered flight (Table 1), the data presumably reflect an unknown mixture of trivial and sustained flights.

### 5.2. Synchrony of Immature Ovaries and Sustained Flight by Females

Evidence that the sustained flights by tethered western corn rootworm were migratory and not simply long appetitive flights is that most of the females making sustained flights had immature ovaries [110,151]. Migratory behavior of many insects occurs during the pre-oviposition period, sometimes concomitant with delayed egg maturation, a phenomenon called the “oogenesis-flight syndrome” [94]. Although many migratory species do not display this syndrome [91,189,190,191,192,193], migratory behavior is strongly suggested when long flights by individuals of a species are characteristically restricted to early in the pre-oviposition period. Female western corn rootworm mate soon after adult emergence, many while still teneral [11,124]. Though sexually mature at emergence in terms of mating competence [122], females emerge with undeveloped, pre-vitellogenic oocytes [152] and have a long pre-oviposition period ranging from 6–21 d [141,150,152,194,195]. Age of peak flight activity in tethered insects is a useful indicator of the migratory window [196]. Western corn rootworm flight mill data (Table 1) support an age of peak sustained flight activity of 3–7 d after emergence, with sustained flight uncommon or rare before and after these ages [151]. Thus, the age of peak migratory activity in this species corresponds to very early stages of ovarian development [61,110,151,152,197] when no or only a few oocytes have left the germarium and entered the vitellarium, and are mostly previtellogenic. Frequency of sustained flight behavior in the lab began to decline as females entered the stage when many oocytes were present in the vitellarium and vitellogenesis had begun, and it ended as yolk deposition accelerated [110,151].

### 5.3. Ascent into Atmosphere for Transport by Wind

The flight speed of migratory insects is usually not great enough to account for observed distances covered in only a few days [66,198,199,200]. Distances of 1000–2000 km traveled by seasonal migrants are not uncommon [201]. For example, migratory flight speed of black cutworm (*Agrotis ipsilon*, Noctuidae), a strong-flying seasonal migrant, averaged about 3.8 km/h (1.06 m/s) on flight mills [202], but a marked male released in Texas took only 2–4 nights to fly 1266 km to Iowa, where it was recaptured in a trap [203]. To accomplish such long displacements in a short amount of time, most migratory insects take advantage of tailwinds [66,80,204]. Wind speeds generally increase with increasing altitude to a maximum at about 200–400 m above ground level (a.g.l.) as strength of the interaction of the atmosphere with the Earth’s surface decreases [205,206]. Migratory flight can occur at low altitudes, especially if the migrant needs to maintain control of its flight direction; diurnal migrants are more likely to employ this strategy [66,200]. However, most migratory insects ascend to high altitudes, up to about 2 km, to facilitate fast downwind transport in winds of higher speed than those available near the ground [80,199,206,207,208,209,210]. Thus, although insect flight near ground level may or may not be migratory, ascent into the atmosphere above the flight boundary layer is a strong indicator of migratory behavior [80].

The reproductive maturity distinctions between migratory and non-migratory western corn rootworms in the study by Coats et al. [61] hold true among field-collected adults flying at different elevations above corn and soybean fields. Western corn rootworm flight activity was monitored [68,167] along with high-elevation flight and ascent around corn and soybean fields in east-central Illinois [58,148,149,211]. In addition to confirming a diel periodicity to canopy-level interfield flight and high-elevation (10 m) flight, analyses of adults collected at 1 m, 3 m, and 10 m a.g.l. reveal a significant increase in the proportion of females among fliers at increasing elevation (0.424, 0.725, and 0.878, respectively) [211]. There were also trends among high-flying (at 10 m) females ascending from field crops and those flying within or just above (at 1 m) the corn or soybean canopies [148]. Females flying at 10 m weighed the least (11.7 × 10^−3^ g), had the lowest percentage carrying any mature eggs (0.6%), and were significantly more likely to contain a spermatophore (84%) [148]. Compared to other females, the characteristics of those flying at 10 m elevation suggest they were mated within the previous 5–7 d. The females flying at 10 m were also likely of local origin as analysis of their gut contents detected the presence of Bt proteins (or other plant tissues) available in upwind source fields [58,211]. Furthermore, the flux (females/min) of beetles active at 10 m elevation rose and fell with variation in the conditions for flight as measured in the canopy of the cornfields surrounding the 10-m platforms used for collection of flying beetles [58]. Thus, most western corn rootworm beetles captured while flying at 10 m had likely just initiated flight from nearby fields and were ascending when they were captured. Their characteristics are similar to those of the migratory individuals identified by Coats et al. [61].

### 5.4. Range Expansion

The historical range expansion of western corn rootworm from its established distribution in the central Great Plains to the East Coast of the United States and southern Canada is well documented (reviewed by [1,4]). Expansion occurred via stratified dispersal [1,4], a process characterized by two scales of spread into previously uncolonized territory [212,213] (see also [214]). Slower spread of the main invasion front can be described as neighborhood diffusion [212], the result of short-range movement driven by station-keeping and local ranging behaviors of individuals, some of whom find themselves in virgin territory. Concurrently, outlying populations are founded by colonizers originating from somewhere at or behind the invasion front. The founder populations have the effect of accelerating the overall range expansion as they spread backward to coalesce with the main front while also spreading forward from their advanced position via diffusion and production of their own long-distance emigrants. Such founder populations were frequently observed during the eastward North American range expansion of western corn rootworm (e.g., [215,216,217,218] and reviewed in detail by Meinke et al. [4]).

Similarly, the western corn rootworm range expansion in Europe, which began after the first detection of an introduced population in 1992 near the Belgrade airport in present-day Serbia [219], has been well documented [16,31,220,221,222,223,224,225]. However, the dynamics of the rootworm’s spread in Europe have a different, more complex flavor than that in North America for several reasons. Analyses of genetic markers indicate there were at least five independent introductions from the U.S. into Europe, which resulted in disjunct areas of infestation [226,227]. Range expansion in Europe has been characterized by spread in multiple directions from multiple disconnected infestations, compared to the more or less unidirectional spread along a broad front that occurred in North America. Numerous disconnected infestations, characteristic of stratified dispersal, have been detected at various, often long distances from the main fronts of diffusive spread in Europe [116,222,228,229].

Distances from the main front of expansion to disconnected outbreaks ahead of the front offer clues about dispersal distances traveled by colonizing western corn rootworm adults. The average rates of expansion, ranging from roughly 20 to 200 km/y in North America [4,215] and 40–100 km/y in Europe [113,224], are logically related to dispersal distances of rootworm adults per generation. The association is not entirely direct, however, making the interpretation of expansion data less than straightforward due to three primary complications.

First, it almost surely takes more than a single mated female to successfully seed a self-sustaining population in a new location [213,230,231,232]. Losses of genetic variation were observed in colonizing populations of western corn rootworm in Europe compared to the likely parent populations [222,224,227,233,234], as is expected from genetic bottlenecks associated with founder events [235,236,237,238]. However, the losses were not of a magnitude suggesting only one or a very few founder females. Regardless, founder events often involve a relatively small number of individuals [213,232,239], and it may take more than one generation for a new population to increase enough to be detectable [240], especially in the absence of systematic monitoring (e.g., rootworm monitoring in the UK pre-2003 [241]). During the time lag between founding and detection, nearby expansion fronts draw closer, so that the minimum distance between a potential parent and founder population at the time of the colonization event is underestimated. Furthermore, if a critical number of immigrants are required to successfully found a new population after settling in close proximity to one another, it seems likely that other individuals disperse beyond that distance but in numbers too few to establish a new population. This consideration is another reason why range expansion data may underestimate the distances western corn rootworms can disperse. Such long-distance movement in small numbers could nevertheless be important for gene flow if the long-distance dispersal event terminates within the already-established larger distribution of the species [89].

Second, the nearest possible source population may not be the actual source. Identification of the most probable source population can be attempted via comparison of genotype profiles between disjunct and potential source populations across an array of selectively neutral genetic markers using population assignment analyses [242,243,244,245,246,247]. This strategy has been employed to determine probable source populations for disjunct western corn rootworm infestations in Europe [222,226,227,234]. Importantly, the closest population is not always the most probable source population based on genetic profiles. For example, the geographically closest possible source population of the Friuli outbreak of western corn rootworm in northeastern Italy was the northwestern Italy infestation, and the nearest possible source of the Frickingen outbreak in Germany was an established population in the Alsace region of France. However, genetic population assignment tests identified the large but more distant Central and Southeastern European (CSE) infestation as the most likely source for both the Friuli and Frickingen disjunct populations [222].

Other types of natural markers can provide evidence for source populations of immigrant insects. For example, genetic and pollen fingerprint analyses were combined with backtrack wind trajectory analyses to identify the most likely source population of boll weevils (*Anthonomus grandis grandis*, Curculionidae) that reinvaded an eradication zone in Texas in large numbers after the passage of a tropical storm [245]. Comparisons of exotic pollen and morphometrics of immigrant black cutworm (*Agrotis ipsilon*, Noctuidae) adults captured near Maryville, Missouri, in the central U.S. in Spring 1985 to those from potential source populations established the Brownsville, Texas area, 1600 km to the south, as the likely source of migrants [248,249]. Similarly, morphometrics of adult western corn rootworm are affected by soil type and other environmental factors and thus differ depending on geographic area, or even field of origin [250,251,252,253,254]. Population genetics analyses could not detect differences between populations within Croatia [233,252,255], but environmentally influenced hindwing morphometrics distinguished four populations within a small 600 km^2^ region in the southeast of the country [252]. Consequently, hindwing shape can potentially serve as a natural biomarker in studies of adult movement and determination of likely source areas of immigrants, including at geographic scales too small for genetic markers to be of much assistance [224,253]. Microbiome communities in western corn rootworm vary by location of sampling across both large and small geographic scales and demonstrate a decay of similarity with geographic distance along transects of populations separated by 12–50 km in northeastern Colorado [256]. The complex interactions between host insects, environment, and means of microbial dispersal outside the host are only beginning to be examined for this genus [257,258]. However, intragenerational comparisons of communities from potential source and receiving populations could perhaps be used as natural markers to identify immigrants, analogous to how species composition of pollen grains attached to an insect’s surface can provide information on its geographic origin [245,259].

Third, it is not always clear whether the founders of a disjunct western corn rootworm population arrived via natural dispersal by flight or by human-mediated transport [222]. After all, at least five different introductions to Europe occurred over the Atlantic Ocean, presumably onboard transcontinental aircraft [226,227]. It is generally assumed that the eastward range expansion in North America was wholly through natural dispersal by flight, an assumption supported by the very long invasion front stretching through areas of the country often far from major air transportation hubs. On the other hand, the U.S. is crisscrossed by major highways, sustaining heavy commercial truck and recreational vehicle traffic, so the role of human-mediated transport in North America cannot be entirely discounted. In Europe, topographical features like the Alps argue against solely long-distance dispersal of rootworms by natural flight from the most logical and genetically supported source population in the CSE infestation region to the disconnected outbreaks of Friuli in northeastern Italy and Frickingen in southern Germany. Carrasco et al. [260] developed a model to help distinguish whether a disconnected outbreak of an invasive insect from an expanding population was most likely from natural flight or from human-mediated transport. They examined western corn rootworm expansion data from Austria to test their model and concluded that many of the disjunct populations ahead of the main front resulted from human transport to the west along the Danube River basin. However, this conclusion is based on assumptions about the ease and frequency of rootworm hitchhiking on boats, trains, and trucks, which so far lack empirical support and may or may not be warranted or more likely than long-distance dispersal by natural flight. Difficulty distinguishing between transport mechanisms (natural flight or human-mediated) in specific cases of founding distant disjunct populations risks underestimating or overestimating western corn rootworm dispersal capacity.

### 5.5. Geographic Spread of Resistance

The rate of spread of insect resistance to a control tactic from a focal population to a new area of previous susceptibility can also provide insight into dispersal distances of the insect. The principle is similar to that of a species range expansion but with the rate of resistance expansion through the existing metapopulation being the measure of the dispersal rate of individuals carrying the resistance allele(s) [40]. The same issues that can complicate inference of flight distances from the rate of the species range expansion also apply to rate of resistance expansion but with additional complicating factors. The most important is that resistance is a phenotype of an individual [46] controlled by one or more genes, each with its own set of alleles, and the resistance phenotype is not a selectively neutral marker.

A well-studied example of using resistance as a marker of western corn rootworm dispersal capacity is the spread of crop rotation resistance. Rotation resistance is manifested as a loss of strict fidelity to cornfields for oviposition. Normally, crop rotation from corn to soybean is a very effective way to control western corn rootworm because the larvae hatching from overwintered eggs deposited in a cornfield cannot survive on soybean roots the following year. In areas with a high frequency of corn–soybean rotation, laying eggs in a soybean field by rotation-resistant females dramatically improves the chance that offspring will hatch in corn the following year. Rotation resistance was first reported in six fields in a 3-km^2^ area of Ford County in eastern Illinois in 1987 [13], from which it spread over the next 15–20 years through most of Illinois and into several neighboring states [170]. Maps of the spread [1,111,112,148,170,261] show a pattern of stratified dispersal with several disjunct pockets of rotation resistance in counties ahead of the main expansion front. Early in the spread of rotation resistance, the annual rate of expansion averaged 10–30 km/generation, with slower rates toward the west and south against prevailing winds than to the east [111,261]. Expansion slowed after 1997 as the front entered regions with higher landscape diversity and possibly higher rates of continuous corn planting [4,112,170]. After the introduction of rootworm-targeting Bt corn into the landscape beginning in 2003, the spread of rotation resistance fully stalled and even contracted [4,262]. Miller and Sappington [40] suggested this could be, in part, a result of widespread planting of rootworm Bt corn even after rotation from soybean, largely negating the selective advantage of rotation resistance. Wetter springtime weather patterns, which significantly reduce populations, have also been hypothesized to contribute to reduced western corn rootworm abundance in soybean fields across areas of Illinois [48]. Regardless, this history of changing rates of rotation resistance spread emphasizes the caution that must be exercised in using spread of resistance, or any other adaptive trait under selection, as a way to estimate dispersal capacity of the insect. Even so, maximum observed rates of resistance spread probably provide a decent estimate in most cases of minimum dispersal capacity.

The spread of cyclodiene insecticide resistance is also instructive. Western corn rootworm resistance to aldrin and heptachlor, used as soil insecticides at planting, was first noticed in 1959 in a small area of south-central Nebraska [263]. By 1963, resistance had spread from that focal area to the eastern front of the species range expansion in western Iowa, southwest Minnesota, and northwest Missouri. It was also spreading in other directions through the existing metapopulation, reaching the western limits of its established distribution in Colorado by 1964 [1,264]. After reaching the eastern boundary of the species distribution in 1963, the subsequent eastward spread of resistance and the species range expansion coincided. At the same time, the annual rate of species range expansion increased greatly compared to preceding decades.

To account for this striking coincidence of resistance development with the accelerated species range expansion, Metcalf [264] suggested that cyclodiene resistance likely resulted, in part, in a behavioral change that caused greater dispersal. While this possibility cannot be discounted, there are several reasons to suspect the increased rate of expansion was due to the phenomenon of stratified dispersal [1,4]. Evolution and spread of cyclodiene resistance occurring at about the same time as the acceleration of the species range expansion is not as surprising a coincidence as it may at first seem, and it does not necessarily signify direct cause and effect. The species’ range expansion out of the western Great Plains was triggered in large part by revolutionary agronomic changes in corn production after World War II, including the use of soil insecticides which played a major role. In other words, the high selection pressure for cyclodiene resistance was intertwined with the same conditions promoting rootworm population growth and eastward range expansion into the Corn Belt: These included increased corn acreage and continuous corn production as a viable option for farmers on the Great Plains, all made possible by efficient irrigation, modern synthetic fertilizer, and use of soil insecticide (cyclodienes) [4]. Until demonstration of increased flight propensity or performance of cyclodiene-resistant rootworms compared to susceptible beetles, e.g., using tethered-flight experiments [185], the most parsimonious explanation for the increased rate of western corn rootworm range expansion starting in the early 1960s is stratified dispersal ahead of the broadening invasion front. Regardless, the rate of spread of cyclodiene resistance within the contemporary range of the species reflects dispersal distances of rootworm adults carrying those resistance alleles [1,264]. Based on distribution maps in Metcalf [264] and Hamilton [265], aldrin resistance spread outward from its origin in the Grand Island and Kearny area of Nebraska in 1959 reaching Brookings, South Dakota, to the northeast by 1963 at a rate of about 80 km/y, and reaching Fort Collins, Colorado, to the west by 1964 at a rate of about 95 km/y.

### 5.6. Spatial Distribution of Resistance

Distributions of adaptive traits in the landscape, including resistance to conventional insecticides and Bt corn, are often quite heterogeneous [169], providing another way to gauge western corn rootworm dispersal patterns and distances. For example, many growers in the western Corn Belt switched to aerial applications of organophosphate insecticides like methyl parathion to control adult western corn rootworm soon after resistance rendered cyclodiene soil insecticides ineffective against the larvae [18,54,56,169]. Surveys of adult resistance to methyl parathion using vial bioassays across several counties in south-central Nebraska from 1995 to 2002 [56,169,266,267] show a patchwork of percentage resistance between and within counties. Most populations in 1996 along a roughly 100-km corridor of susceptibility between highly resistant populations in Phelps Co. and populations of building resistance in York Co. to the northeast had become highly resistant by 1998 [169]. When resistance appears to spread from concentrated sources like this, it suggests geographic spread of resistance alleles by dispersing adults. However, resistance to methyl parathion or other control measures does not behave like a selectively neutral marker; thus, local selection, while high in many areas, was low or non-existent in others. Reinders et al. [34] listed some factors influencing the level of Bt resistance of rootworms in and between fields at the local landscape scale. With modification, these principles apply to spatial variation in resistance to conventional insecticides and rotation resistance as well, including how long the control measure has been used, population density in the area, rates of gene flow with distance (a function of adult dispersal), and frequency of alternative control tactics including efforts to mitigate resistance. If resistance alleles are already present in a susceptible population, the rate at which the phenotype is manifested depends in part on local selection pressure, rate of introduction of resistance and susceptible alleles carried by immigrants, and population density [34].

As with the spread of rotation resistance, it is problematic to directly translate rates of spread of methyl parathion resistance phenotype into estimates of rootworm dispersal distances. Nevertheless, some inferences about western corn rootworm adult dispersal can be gleaned from such data. The presence of a susceptible population in the near vicinity of one or more resistant populations indicates that the immigration rate of the resistant insects was not sufficient to overcome susceptibility. For example, 8 of 11 populations surveyed in 1996 in Phelps Co., Nebraska, were highly resistant to methyl parathion, 3 of them located only 5–7 km to the west, south, and east of a susceptible population [169]. Populations in Gosper and Buffalo Counties susceptible to methyl parathion in 1995 [56] had become resistant by 2002, but this was probably facilitated by selection from this insecticide used commonly in those counties [267]. However, a formerly susceptible population in central Clay Co. (as of 1998 [169]) was also resistant by 2002 [267] despite lack of selection pressure from methyl parathion, indicating that the increase in resistance must have been spread by resistant immigrants. If a nearby population was the source of resistance alleles, the closest known resistant population in time and space (1999, northwest Clay Co. [266]) was ~30 km away, yielding a dispersal estimate of about 8 km/y. Because many more than one resistant beetle would be needed to shift the susceptible population to resistant, 8 km/y can only be considered a minimum estimate of dispersal distance by the resistant adults. Tethered-flight experiments [185] indicated flight activity of methyl parathion-resistant adults did not differ from that of susceptible adults, although exposure to the insecticide reduced flight activity of resistant beetles. Duration of sustained flights (i.e., >20 min) among unexposed resistant rootworm adults averaged 52.7 min. Given average speeds for sustained flights of western corn rootworm adults measured on rotary flight mills of 50 m/min (0.83 m/s) [61] and 28.8 m/min (0.48 m/s) (archived data in [100] https://figshare.com/s/8f7b424145e5ac9fb8e9?file=14362775, accessed on 26 November 2023), methyl parathion-resistant adults could be predicted to fly 1.5–2.6 km per sustained flight. Expected total distances covered by individuals would be greater if aided by wind, or if more than one sustained flight is taken by a migratory adult along the same trajectory.

Another interesting example involves development of resistance to carbaryl, the active ingredient in an insecticidal bait used in a pilot areawide management program from 1997 to 2001 [268,269]. In this pilot program, fields in management areas of ~41 km^2^ located in four Corn Belt states were treated with a cucurbitacin bait mixed with the insecticide carbaryl to control adults [270]. Cucurbitacin is a semiochemical arrestant and feeding stimulant for western corn rootworm adults [271,272,273]. Pruess et al. [53] had previously shown that areawide adult control with malathion across a block of this size could successfully protect treated fields from economic injury by larvae the following year, but that cumulative suppression over years was not achieved because of recolonization by adult immigrants the year after treatment. Similar results were observed in the areawide cucurbitacin bait study [18,274]. In three of the four states, resistance to carbaryl increased significantly within the managed block during the period of the study compared to nearby untreated control fields [269]. There was also a trend of decreasing susceptibility in the nearby control fields, suggesting possible spread of resistance from the management block, but the change was subtle and not statistically significant [268,269]. In addition, there was a significant decrease in behavioral response (as arrestant, feeding stimulant, or both) to cucurbitacin in the managed areas compared to the control fields [269]. Together, these results imply that immigration of surrounding susceptible western corn rootworm adults into an area of 41 km^2^ was not enough to overcome localized selection for resistance, and that emigration out of the managed area was not great enough to significantly increase resistance in nearby susceptible populations. At the same time, immigration into the managed area was enough to prevent cumulative population suppression.

Rootworm Bt corn was rapidly embraced by growers throughout most corn-growing regions east of the Rocky Mountains after its commercial release in 2003. It provided excellent control of rootworm larvae, allowing farmers to plant continuous corn while forgoing the use of chemical insecticides. Unfortunately, the western corn rootworm lived up to its reputation for quickly surmounting control tactics through evolution of resistance, in this case to the various Cry3 toxins expressed in the Bt-corn hybrids [37,38,43,275,276,277,278,279,280]. Resistance to the different Cry3 toxins (Cry3Bb1, mCry3A, and eCry3.1Ab) is now widespread, thanks in part to extensive cross-resistance among the structurally similar toxins [37,38,276,277,281,282], but was initially patchy [43]. Reports of field-evolved resistance to Cry34Ab1/35Ab1 (hereafter “Cry34/35”; note this toxin was recently renamed Gpp34Ab1/Tpp35Ab1 [283]) Bt corn [275,278] emerged after an increase in its adoption in response to Cry3 resistance [284]. Like early resistance to Cry3 hybrids, the geographic distribution of Cry34/35 resistance is patchy [275,280].

For both Cry3 and Cry34/35 toxins, the initial appearance of resistance in widely separated locations, as well as cases of co-occurrence of resistant populations with nearby susceptible populations (e.g., [34,38,43]), suggest multiple instances of independent evolution of resistance in local selection “hotspots” [37,38,40,43,46]. The existence of resistance hotspots is made possible by the large proportion of resident beetles comprising an adult western corn rootworm population [19]. Adaptive alleles can build up over time in response to strong selection on a (mostly) sedentary population of residents that oviposit most or all of their eggs in the natal field or surrounding cluster of fields comprising the natal home range. As described earlier, however, not all residents remain in the natal home range. Depending on environmental conditions, some residents may range via appetitive flights in search of more attractive habitat, depositing a portion of their eggs in fields in the local landscape but outside the natal home range (Figure 1). The migrant portion of the population migrates after mating but before ovipositing, ultimately leaving their genes in fields potentially many kilometers away. The spatial distribution of resistance to Bt-corn in the landscape around a hotspot reflects the outward rate of resistance spread, or expansion through the surrounding metapopulation, and thus may be informative about adult dispersal rates and distances.

Gassmann et al. [37] concluded that despite the relative proximity of fields in northeastern Iowa that developed resistance to Cry3 Bt-corn hybrids by 2009 and 2010, field histories of at least 3 consecutive years of Bt-corn planting (the minimum necessary for resistance to evolve in a population [34,38,41,43,285,286]) were consistent with independent evolution of resistance. However, in one of these fields, the resistant rootworm population received only one year of Cry3 selection, indicating that the field must have been colonized by Cry3-resistant adults. Resistant populations were present in the same and adjacent counties, but the distance from which the resistant colonizers originated is not known.

St. Clair et al. [36] compared field histories, larval damage, and population sizes in “problem” counties in northeastern Iowa, where greater-than-expected injury (i.e., >1 on the Oleson et al. [287] node injury scale) to Bt cornfields had been reported previously, to “non-problem” counties in southeastern and central Iowa where there had been no such reports. Planting of continuous corn was common in the surveyed fields of all counties and years (2015–2017) examined. Surprisingly, larval damage, abundance, and resistance to Cry3Bb1 Bt-corn did not differ between these counties. The authors concluded this was because mitigation tactics quickly adopted in the problem counties, such as soil insecticide applications and switching to Cry34/35 Bt hybrids, successfully suppressed the Cry3Bb1-resistant rootworm populations. The temporal and spatial patterns of the widespread Cry3Bb1 Bt-corn resistance suggested independent selection for resistance dependent on differing selection pressures in the different regions of the state (e.g., less intensive maize production in central and southwestern Iowa than in the northeast) rather than spread of resistance from northeastern Iowa where field failures were first observed. In a companion study, St. Clair et al. [288] made similar comparisons at the local landscape scale, between focal fields with at least one year in the previous six of greater-than-expected damage to Cry3Bb1 or mCry3A Bt-maize and neighboring fields (within 2.2 km). Again there was no difference in injury or rootworm abundance, and populations from all fields were resistant to Cry3Bb1. The focal fields, which had suffered greater-than-expected damage to Cry3Bb1 maize in the past, had a field history of more years of planting Cry3 maize than surrounding fields. While it is clear that resistance alleles were spread from the focal fields to neighboring fields in the local landscape, the authors pointed out that field-level selection was also occurring in the surrounding fields. They suggested that immigration of resistant beetles from a focal field augments the effect of selection in receiving fields and thus helps homogenize the level of resistance in the landscape.

In a fine-grained geographic study of 17 fields in a 10 × 20 km area of Keith County, Nebraska, and 16 fields in an 11 × 13 km area of Buffalo County, Reinders et al. [34] documented a mosaic of Cry3Bb1 and mCry3A resistance and susceptibility in the landscape. They found multiple instances of fields harboring Cry3 Bt-resistant western corn rootworm populations within 1–3 km of fields with susceptible populations. The authors classified each field with a cumulative index score of selection pressure from Cry3 Bt corn derived from field history (e.g., number of years per trait, single or pyramided traits, number of resistant or susceptible populations within 1 mile, etc., with the index reset to 0 after crop rotation to soybean). By comparing selection index scores with observed levels of Bt resistance, inferences about gene flow of both susceptible and resistance alleles were possible in some cases. As in the Iowa case described by Gassmann et al. [37], Reinders et al. [34] documented substantial Bt resistance among larvae from second-year corn after rotating out of soybeans, indicating they must have been offspring of Bt-resistant immigrants arriving from elsewhere during the previous year. This field (Field 4) was less than 1 km from the border of the sampling area in Buffalo County, and the authors suspected the immigrants came from fields of continuous Cry3Bb1 Bt corn located “immediately adjacent” to Field 4 but outside the surveyed area, and thus they were not bioassayed for resistance.

### 5.7. Gene Flow

Because of its intimate relationship with dispersal, estimates of gene flow can, in principle, be used to infer distances and patterns of dispersal using a variety of population genetics approaches [246,247,289,290,291]. The most common methods use variation in allele frequency across panels of selectively neutral genetic markers to quantify genetic differentiation between sampled populations [292,293,294,295]. *F*_ST_ is a measure of genetic differentiation between populations or subpopulations, ranging from 0 (no difference in allele frequency) to 1 (fixed differences in alleles) [296,297]. Because genetic differentiation of two populations is inversely related to the rate of gene flow between them, a rough estimate of migrant exchange per generation, *Nm*, can be calculated directly from *F*_ST_ values using the equation *Nm* = [(1/*F*_ST_) − 1]/4 [298]. In areas where individuals are distributed continuously across the landscape, one expects to see genetic differentiation increase with geographic distance, creating an isolation-by-distance (IBD) relationship [297,299,300]. IBD is assessed via regression of genetic distance, *F*_ST_/(1 − *F*_ST_), on geographic distance [297,301,302]. It reflects the balance between gene flow (determined by effective dispersal distances) and genetic drift (a function of effective population size).

While theoretically sound and intuitively appealing, use of *F*_ST_ to infer gene flow (and thus dispersal) relies on several critical assumptions: (1) The genetic markers are selectively neutral; (2) mutation rate of the markers is relatively low and constant; and (3) the sampled populations are at gene flow–genetic drift equilibrium [303,304,305]. Violation of any of these can affect *F*_ST_, in which case geographic patterns in degree of differentiation cannot be ascribed unambiguously to gene flow [291,306].

A number of researchers have employed population genetics strategies to help address uncertainties related to the movement ecology of adult western corn rootworm and have reported pairwise *F*_ST_ values (Table 2). For example, Kim and Sappington [307] estimated genetic differentiation between 10 populations from New York to northwestern Texas using microsatellite markers. None of the pairwise *F*_ST_ values were significant except those involving Texas samples, and two involving the Illinois sample versus the Pennsylvania and Delaware samples. The IBD regression was significant but only when the Texas population was included, a region of probable hybridization with the partially reproductively isolated Mexican corn rootworm (*Diabrotica virgifera zeae*) [308,309]. Flagel et al. [310] used a large number of transcriptome-derived SNP markers to examine differentiation among 20 wild populations of western corn rootworm from northeastern Colorado to eastern Indiana. Pairwise *F*_ST_ values between populations were calculated for all (>10,000) unigenes identified from the transcriptome. The mean *F*_ST_ between all populations was 0.052 and indicated little differentiation even at great distances, up to 1500 km. IBD was detected across the sampled area but only when the easternmost population sampled, Indiana, was included. The reason for this outsized influence of the Indiana population is unclear [310].

Likewise, other studies consistently report low and mostly nonsignificant genetic differentiation among western corn rootworm populations through the vast U.S. Corn Belt (Table 2) (see [18] for a map of the Corn Belt). Although such findings can theoretically reflect dispersal and gene flow over very long distances, the various authors recognized the more likely explanation is that gene flow–genetic drift equilibrium has not yet been attained in the wake of the recent eastward range expansion [1]. This interpretation is supported by studies in which populations south of the Corn Belt, such as Mexico, Arizona, New Mexico, and Texas, were included in comparisons (e.g., [227,311,312]). The high and significant pairwise *F*_ST_ values reported in comparisons among themselves and with northern populations (Table 2) reflect the much longer period of western corn rootworm residency in southwestern North America than in the Corn Belt [312].

**Table 2 insects-14-00922-t002:** Summary of pairwise *F*_ST_ estimates between western corn rootworm populations reported in the literature.

		Geographic Scale (km)							
Region	No. Sites	Min	Max	Genetic Markers	Signif. IBD?	Pairwise *F*_ST_ Range ^a^	% Signif. *F*_ST_ Values	Notes	Citation
USA:	10 (9 states)	9	1400	Microsats	Yes (No when TX excluded)	0–0.002	24 (6 when TX excluded)	Texas sample from a probable hybrid zone with *D. v. zeae* (Mexican corn rootworm).	Kim and Sappington [307] (See also Kim et al. [244] and Coates et al. [313] for similar results using EST-derived microsat and SNP markers, respectively.)
NY to									
northwest TX									
USA:				Microsats and AFLPs	—			Sampled pupae instead of adults. IL populations were rotation-resistant; IA populations were wild-type.	Miller et al. [314]
central IA	3	3	32			IA: 0.007	IA: 0		
east-central IL	3	0.5	1			IL: 0	IL: 0		
IA vs. IL	2	500	500			IA vs. IL: 0.002_AFLP_ to 0.005_Microsat_	IA vs. IL: 100		
USA:	4	254	1941	Microsats	—	0–0.004	0	Several lab colonies were also compared but not included here.	Kim et al. [315]
PA, IA,									
eastern KS,									
western KS									
North America:				Microsats	—			For European comparisons, only populations within a single outbreak area are included here.	Ciosi et al. [227]
PA, IL, TX, AZ,	5	763	3124			0.009–0.118	100		
Mexico									
Europe:						0–0.023			
northwestern Italy	3	21	40				67		
Europe:				Microsats				Two locations were shared between the East and West transects.	Ciosi et al. [222]
CSE (total)	18	14	438		Yes	0–0.039	13		
East transect	11	24	433		Yes	0–0.017	2		
West transect	9	26	438		No	0–0.039	13		
Europe:				Microsats	—			Only comparisons between populations within outbreak areas, and not admixed, are included here.	Bermond et al. [228]
northwestern Italy	3	73	151			0–0.01	33		
CSE (Hungary, Serbia)	2	211	211			0	0		
USA:	5	9	1400	Microsats	Yes (No when AZ excluded)	0.002–0.082	80	2 NE populations were insecticide-resistant (R), and 2 (+AZ) were susceptible (S). All comparisons were significant, except within R or S categories in NE.	Chen et al. [311]
eastern NE, AZ									
Europe:				Microsats				In 2009, 10 of 11 sites were within 80 km of each other in eastern Croatia, and all were ≥245 km from Ogulin at the western edge of the invasion front.	Lemic et al. [233] (See also Lemic et al. [234,252] for similar results including samples from 2011.)
Croatia 1996	11	7	182		No	0–0.083	15		
2009	11	7	393		Yes (No when Ogulin excluded)	0–0.024	4		
USA:	20 (8 states)	0.5	1528	SNPs (transcript-omic)	Yes (No when IN excluded)	0.029–0.054	—	Six laboratory colonies of differing years in culture and originating from various locations were also compared, but are not included here.	Flagel et al. [310]
Corn Belt									
(CO to IN)									
USA:				Microsats	—			The 3 populations from IL were rotation-resistant.	Ivkosic et al. [255]
NE, IA, IL	7	110	800			0–0.005	0		
Europe:									
Croatia, Serbia	8	15	350			0–0.011	0		
North America: Mexico to Northeast USA	21	92	3456	Microsats	Yes	0–0.160	65 (31 from KS to north and east)	Neighbor-joining tree indicated tight clustering of sites north and east of KS.	Lombaert et al. [312]

IBD, isolation by distance. Microsats, microsatellites; AFLPs, amplified fragment length polymorphisms; SNPs, single-nucleotide polymorphisms. USA state abbreviations: AZ, Arizona; CO, Colorado; IA, Iowa; IL, Illinois; IN, Indiana; KS, Kansas; NE, Nebraska; NY, New York; PA, Pennsylvania; TX, Texas. CSE, Central and Southeastern European outbreak area. ^a^ Negative *F*_ST_ estimates in original papers are reported here as ‘0’.

In Europe, high and significant pairwise *F*_ST_ values are typically reported between geographically distinct regions of infestation. These disjunct populations either originated directly from independent introductions from North America or via secondary emigration from invasive bridgehead populations [316]; the observed genetic differentiation undoubtedly reflects founder effects associated with recent colonizations [222,226,227,228,234]. In contrast, differentiation within larger areas of infestation that originated from a single founder population is invariably low and mostly non-significant (Table 2) [222,226,227,228,233,234,255].

The lesson is that *F*_ST_-based inference of western corn rootworm dispersal distances is unreliable for populations in areas of recent invasion, such as the U.S. Corn Belt and Europe because the assumption of gene flow–genetic drift equilibrium is violated. However, it should be possible to use patterns of genetic differentiation to infer dispersal distances in regions of long-term endemicity, such as in the western Great Plains. One such region is northeastern Colorado, which served as the source of the eastward range expansion. Based on molecular genetic analyses, Colorado was colonized in ~1838 CE by western corn rootworm originating from populations further south in New Mexico and Texas [312]. Pairwise *F*_ST_ values between northeastern Colorado and all locations further east in the Corn Belt indicated significant differentiation (Table S3 in Lombaert et al. [312]). Preliminary analyses among populations sampled along transects within northeastern Colorado indicate significant IBD as well (T.W.S. and J.L.S., unpublished data), as would be expected in an area at gene flow–genetic drift equilibrium.

Even in areas of invasion, however, other types of genetic analyses can be exploited to provide estimates of dispersal. For example, Bermond et al. [62] estimated dispersal distances of western corn rootworm by exploiting distinct genetic profiles at microsatellite marker loci of independently introduced and expanding CSE and NW populations in Europe as they came in contact in northeastern Italy. The principle behind their estimates is that transient clines of selectively neutral genetic markers will form through gene flow in the zone of contact (equivalent to a hybrid zone) when the expansion fronts of differentiated populations meet. The decay rate of the cline as genotypes homogenize via interbreeding, which is measured by change in the width of the contact zone, depends on the diffusion rate of dispersal [317]. In addition, Bermond et al. [62] calculated dispersal distance from the width of the linkage disequilibrium cline generated by gene flow between the two expanding populations, which is maximum in the center of the contact zone [318]. Estimates from both methods were in broad agreement, suggesting western corn rootworm dispersal of 13–21 km/generation. This is not only strong evidence for relatively long-distance movement by western corn rootworm adults, but it also presents a clear example of *Slatkin’s paradox*, a dilemma in which estimates of gene flow suggest movement over much greater distances than the much shorter lifetime dispersal distances indicated by direct ecological studies [27,28,304]. As pointed out by Bermond et al. [62], the cline analysis estimate of per-generation dispersal is the same as the diffusion rate but only if long-distance dispersal is rare. It is likely, given ecological evidence of less-common, but not rare, long-distance movement by western corn rootworm, that the estimate of 13–21 km/generation from clinal decay analysis reflects both short-distance diffusion via station-keeping and ranging flights as well as long-distance leaps via migratory flight.

## 6. Conclusions and Synthesis

As summarized in this review, a wide array of experimental and observational data strongly supports short-distance lifetime displacement by western corn rootworm adults. This deduction starkly contrasts with evidence strongly supporting long-distance lifetime displacement from an equally imposing array of data from indirect ecological, behavioral, and population genetics studies. The geographic scale of estimated net lifetime displacement of western corn rootworm ranges from a few tens of meters to hundreds of kilometers (Figure 2). This lack of congruence in estimated dispersal distances across methodologies is a classic example of both Reid’s and Slatkin’s paradoxes. As profound and confounding as these contradictory estimates are, they are not illusory or generated by methodological shortcomings. The weight of evidence is clear that both short- and long-distance flights occur in this species, generating a population-level pattern of bimodal dispersal distances. However, the outstanding question concerns the mechanisms generating this pattern.

In other species, both paradoxes are most commonly resolved by positing rare, unobserved, or underestimated long-distance dispersal events by a portion of the population [24,27]. In other words, it is assumed that essentially “invisible” long-distance movement (invisible because it is rare and difficult to observe) comprises the long tail of a leptokurtic frequency distribution of dispersal distances in a population [22,27,28,319]. As is true of frequency distributions for measures of dispersal of most organisms, including insects [196,214,320,321,322], the frequency distributions of flight duration of western corn rootworm observed in tethered-flight experiments are always positively skewed and leptokurtic, with many short bouts of flight tapering off to a long tail of progressively less-frequent longer flights [61,100,110,151,183,185]. In principle, the long leptokurtic tail could simply represent facultative, long-distance ranging flights by individuals in search of better habitat. If this were the case, however, one would expect a “thin”-tailed frequency distribution of distances typical of a normal or Gaussian distribution, where long-distance flights quickly become very rare [321,322]. This is the way stratified dispersal has been conventionally modeled [323]. Instead, the evidence for western corn rootworm indicates that very long flights >10 km, while certainly much less common than short-distance flights, are not exceptionally rare. The resulting pattern of a fat-tailed distribution of dispersal distances is evident in the tethered-flight experiments and is inferred from population genetics studies [62] and observations of disjunct founder populations far ahead of invasion fronts [4,222].

In this paper, we have reviewed the different lines of evidence accumulated over many decades for both short- and long-distance displacement of western corn rootworm, including what is known about this insect’s flight behavior, such as temporal and developmental timing and the underlying motivations for bouts of movement. Based on these many studies and their findings, we conclude that two behavioral phenotypes, residents and migrants, exist in any western corn rootworm population. Individuals of both phenotypes engage overwhelmingly in appetitive flights throughout their life, but only one phenotype also engages in non-appetitive, migratory flight. We propose a conceptual model of western corn rootworm as a partially migratory species, where the resident portion of the population never engages in migratory behavior, while the migrant portion does (Figure 1 and Figure 2). Thus, the observed “fat” leptokurtic tail of flight distance is shaped by a mixture of two dispersal kernels [22,25,322], one representing the frequency distribution of short-distance appetitive flights of both phenotypes and the other the distribution of long-distance non-appetitive flights by the migrant portion of the population. The presence of two behavioral phenotypes, manifesting as two dispersal kernels emerging from the same western corn rootworm populations, is the solution to both Reid’s and Slatkin’s paradoxes in this species.

Whether long-distance movement by western corn rootworm is via appetitive extended-ranging flight or non-appetitive migratory flight is not an exercise in semantics. It affects how experimental data on movement are interpreted and applied to predict population dynamics, efficacy of management strategies, and evolution of local adaptations like resistance. For example, emigrants from a field are often equated with the ~15% of tethered females across all ages tested by Coats et al. [61] that engaged in migratory flight. Consequently, this percentage is commonly used in models or interpretations of field data as the expected rate of emigration from a natal field. However, not all emigrants from the natal field are migrants. Additionally, not all immigrants entering a field are migrants. Individuals that leave a field and enter a different field may be residents engaging in simple appetitive station-keeping flights or ranging flights (Figure 1). The flight mill studies of western corn rootworm by Coats et al. [61] and Naranjo [110] only distinguished migrants from non-migrants. But “non-migrant” is a descriptive category, not a phenotype. A group of non-migrant rootworm adults observed on a particular day are a mixture of residents (which never migrate) and migrants that did not happen to engage in migratory flight on the day of observation. Residents emigrate from or immigrate to fields only via appetitive flight, even if they have traveled far enough to exit their previous home range. The point is that individuals of the migrant phenotype constitute only a subset of the emigrants from and immigrants to a given field, such as the natal field. The total number of adults emigrating from a field will be greater than the number of migrants emigrating because a collection of emigrants will include individuals of both migrant and resident phenotypes. Similarly, the number of all immigrants contributing to a field’s total population will be greater than the number of migrants that entered and now comprise part of that field’s total population. Thus, when a % value of female migrants observed in flight mill studies, like the 15% value of Coats et al. [61], is used to parameterize emigration rates from the natal field, it will underestimate actual emigration. This is because the 15% value only includes migrants and does not account for emigration via appetitive flight, like ranging behavior or even station-keeping behavior that spills over boundaries of adjacent fields. In addition, the percentage of migrant females in a population is probably closer to 25–30% (potentially up to 50%), based on tethered-flight data when restricted to females under 10 d old (Table 1), the ages encompassing the developmental window for pre-oviposition migratory behavior [58,148,149,211].

Immigrant western corn rootworm adults will include a subset of erstwhile migrants that henceforth behave like residents in their new home range (Figure 1). Any future emigration events the immigrants engage in probably will not involve migratory behavior but rather appetitive flights, which could include ranging over relatively long distances. An exception could be multi-leg migration flights taken in short succession by an individual still in a stage of early ovarian development, but whether an adult engages in more than one migratory flight on different days is unknown. Females in tethered-flight experiments that made a migratory flight, sometimes made more than one in the same 24-h test period [61,183]. However, direct application of laboratory-derived flight parameters may overlook real-world constraints on flight, a well-known issue with all tethered-flight studies [196,324,325,326]. For example, in the study by Coats et al. [61], some females engaged in multiple overnight flights. This is unlikely under field conditions. Isard et al. [58] showed that among individuals in field populations, ascending flight stops at or just before sunset. Thus, a long-distance migrant that stops flying and alights after dark will not be able to initiate another flight until the return of daylight and the presence of permissive atmospheric conditions [58,167].

## 7. Unanswered Questions and Future Directions

### 7.1. Rate of Emigration from the Natal Field

Characterizing the relative contributions of station-keeping, ranging, and migratory types of flight behavior to emigration and immigration rates of western corn rootworm will require different approaches to obtain the data needed to parameterize adult movement functions in models. Although the 15% value commonly used for lifetime emigration rate is almost certainly a substantial underestimation as explained above, determining a more realistic rate may not be simple. The true maximum rate of emigration must be considerably less than 100% because we know that enough females lay eggs in their natal field to cause the observed population increases over consecutive generations in continuously planted cornfields. At the same time, the true rate of emigration must be substantially greater than nil, because those of the migrant phenotype will always emigrate (unless prevented over an extended period by prohibitive abiotic conditions), and because we know that first-year cornfields are rapidly colonized by immigrants, presumably of both phenotypes.

A focus on measuring rate of emigration directly from an isolated field could be an effective strategy. However, determining the rate or ceiling rate of emigration may be difficult. One possibility would be to compare the observed rate of population increase in a focal field from one generation to another to the rate of predicted population increase in the absence of emigration and immigration, to calculate the net deficit of adults in the field at a given time. This is a question of population dynamics that lends itself to modeling. Contributors to the deficit will include adult mortality and the difference between immigration and emigration. The challenge lies in disentangling these three factors [327,328]. This approach is similar to that taken by Hein et al. [327] to quantify the relationship between population density of western corn rootworm adults and oviposition in cornfields. They obtained direct estimates of adult emergence with emergence cages and estimated adult population densities over the season via whole-plant visual counts [329]. Early in the period of adult emergence, there was good correspondence between rate of emergence and number of adults in the natal field. However, within three weeks after initial emergence, only a third of the expected beetles were present in the field, indicating either high mortality, high emigration, or some combination of the two. They recognized immigration could also have affected population density but assumed for their model that it was less than the rate of emigration; this was a reasonable assumption because the fields they studied were not late-planted and should not have been more attractive to would-be immigrants than other fields in the landscape.

### 7.2. Mechanisms Determining Resident and Migrant Phenotypes

In most partially migratory insect species [67], the underlying mechanisms determining the resident and migrant phenotypes are poorly understood. In general, there is a strong polygenic basis for migratory syndromes and component traits in insects [64]. It can also be presumed that even though the resident and migrant phenotypes in western corn rootworm are distinct dichotomous traits, it is likely that the final phenotype is not determined by a single major gene but by a threshold response [91,330,331,332] to a continuous distribution of many genes of small effect. This kind of situation manifests as a “developmental switch” mechanism: a migrant phenotype develops when enough genes with alleles associated with migration are present to exceed a threshold, while a resident phenotype develops when the threshold is not exceeded [67,333,334].

At the same time, the ultimate phenotype of partially migratory insects often depends on environmental interactions with the various genes associated with the behavioral phenotypes [67,91,333]. Environmental cues, often experienced by an immature stage during a sensitive period, influence the decision to develop as one or another phenotype [335]. The lag period between the environment-sensitive period in the immature stage and production of an adult migrant may be needed to allow time for development of the associated morphological and physiological components of the species’ migratory syndrome. For example, in the partially migratory oriental armyworm (*Mythimna separata*; Noctuidae) and beet webworm (*Loxostege sticticalis*; Pyralidae), the decision to develop as a migrant or resident adult occurs in the larval stage. Environmental cues received as larvae, including poor nutrition, short photoperiod, cold stress, and larval crowding, lead to development of the migrant adult phenotype. Environmental conditions encountered during the first day of adulthood can shift an erstwhile migrant to a resident, but an adult resident cannot shift to become a migrant [81,336,337,338,339].

The western corn rootworm has not been studied as extensively in this regard, but there are indications of underlying genetic variation in flight behavior. For example, Li et al. [251] examined the effects of fields and regional populations of origin, and the number of generations of laboratory rearing on flight activity of offspring at 1–6 d of age under identical rearing and assay conditions. Significant differences in the means and variability of propensity to take off and seconds to takeoff from a vertical stick were common. Larvae reared at high density are more apt to fly and fly farther on flight mills than those reared under less crowded conditions [100], suggesting crowding may be an environmental cue affecting adult phenotype. Similarly, in the paired-field study by Levay et al. [52] described above, the percentage of immigration (and hence emigration under the assumptions of the experimental design) increased with increasing estimates of adult population density.

It is likely that most or all developmental control mechanisms underlying partial migration in insects include hormonal signaling [335,340], and juvenile hormone (JH) is commonly involved (e.g., [64,196,337,341,342,343]). Importantly, Coats et al. [151] demonstrated that JH applied to the adult western corn rootworm cuticle increased the likelihood of migratory flight on flight mills. This finding does not necessarily indicate the adult stage is the environment-sensitive period in rootworms for phenotype determination because environmental cues received by larvae can trigger effects on JH concentrations, timing of JH secretion or sensitive period, threshold sensitivity, or cellular responses much later in development [335]. There is great scope for future research focusing on the genetic and environmental factors and their interactions that determine the developmental trajectory of individual western corn rootworms into resident or migrant phenotypes.

### 7.3. Differential Migration of Males and Females

Evidence from atmospheric ascent and tethered-flight data suggests the percentage of males that migrate is much lower than that of females. The proportion of females among western corn rootworm captured 4.6–10.0 m a.g.l. consistently ranges between 72% and 89% [58,149,211,344]. Of the beetles washed up on the shores of Lake Michigan over a three-year period, an average of 89% were female. Although tethered flight studies have tended to focus on females, available data indicate only 1–11% of tethered males made a sustained flight [110,183,184,185], in sharp contrast to the 21–54% observed for females (Table 1). Although these same data support migratory flight by some males, the substantial differences suggest sex-dependent differences in costs and benefits of migration. Male genes migrate whether males do or not because most migrant females mate before departing. Thus, any advantage to the male of physically migrating may be reduced and its risks amplified. By migrating, a male could potentially improve its mating chances, but perhaps little more than by ranging in the local landscape. Conversely, leaving a landscape where receptive females are currently plentiful to alight where the presence of unmated females is not assured poses a higher level of risk than faced by a mated, migrant female that only needs to find a cornfield in which to feed and oviposit. This type of risk is elevated during range expansion, when a migrating male that leaps ahead of the invasion front will encounter a new landscape essentially devoid of young calling females. If, as seems likely, a quantitative genetic developmental switch is involved [67,91] in determining resident or migrant phenotypes, the observed female bias among migrating adults implies a higher “migrant” threshold response in males than in females. Regardless, observed sex-dependent differences in propensity to migrate potentially complicate parameterization of the movement components of IRM and population models, where male dispersal affects spatial encounters and mating with newly emerged females (e.g., [105]).

### 7.4. Migration Process

#### 7.4.1. Ascent Phase

The three phases of migration (ascent, transmigration, and termination) are inherently difficult to observe and characterize for any migratory species [66,206,345], and much is unknown about the behavior of individual western corn rootworm during migratory flight. There are two diurnal peaks of western corn rootworm ascent to at least 10 m a.g.l. in east-central Illinois, one in the morning between about 06:45 and 11:00, and the other in the evening from 17:00 to 20:30, ending just before sunset [58]. Flight mill data generally support morning and evening peaks of sustained flight activity [61,100]. In the study by Coats et al. [61], sustained flight of females was most common from 16:00 to 22:00, with a smaller peak from 05:00 to 06:00, and a few instances between 23:00 and 03:00. However, the entire 24-h test period of this experiment was conducted under constant dim lighting to simulate twilight, making translation of timing in the lab to that in the field questionable. Flight mill trials in the study by Yu et al. [100] were conducted under a 14:10 L:D photoperiod with simulated dawn and dusk. The longest sustained flights (≥10 min) were initiated by 75% of individual females between 07:00 and 15:00 (median 08:30). However, 40% of the longest flights were initiated at first-light at 07:00, suggesting a startle response, again making translation to field timing suspect. Regardless, none of the longest sustained flights started before the beginning of dawn, and all ended before sundown.

Dissections of female western corn rootworm captured at 10 m a.g.l. while ascending from corn revealed 99% were mated, 84% with spermatophores, and half of the spermatophores were large, indicating very recent mating [148]. It seems unlikely that this tight tie between mating and initiation of migration is coincidental. One possibility is that the spermatophore helps supplement energy requirements during migratory flight. Another is that mating stimulates sustained flight in the migrant phenotype.

If the potential for migratory flight is enabled by mating, perhaps flight mill trials using females from interrupted matings, like those studied by Sherwood and Levine [141], could yield insight into the role of mating (or mating quality) on initiation of migratory flight. Sherwood and Levine [141] used interrupted western corn rootworm mating treatments to show that a copulation duration (1 h) insufficient for sperm to be transferred into the spermatheca resulted in a lower percentage of egg-laying females and few eggs laid. Their study strongly suggests that the mechanical stimulation associated with copulation or fecundity-enhancing substances (FESs) produced by male accessory glands is responsible for eliciting the mated response in females. FESs are transferred to the female during mating, presumably as spermatophore components. Alternatively, in the many species in which FESs are implicated in the transition to mated-behavior patterns (i.e., cessation of mating receptivity, initiation of egg maturation and oviposition), including some beetles, hemocoelic injection of male reproductive tract homogenates alone is sufficient to change female behavior [346,347,348,349,350,351]. Hemocoel injection of homogenized WCR male accessory glands (which are the source of spermatophore components) or whole spermatophores themselves (recovered from freshly mated and incompletely mated females) into unmated females destined for flight mills would present another route to examine the male contribution on subsequent expression of migratory behavior—free from the potential influence of mechanical stimulation.

Though the western corn rootworm is clearly diurnal as an adult, the extent to which they may also fly at night is unclear. Collections on sticky traps in cornfields at heights up to 3.6 m showed morning and afternoon peaks of flight activity for both sexes, but some beetles were captured during the 8-h period of darkness [126,168]. Tóth et al. [352] found that male western corn rootworm responded to pheromone lures throughout the night. Laboratory tethered-flight studies under light–dark cycles show substantial short-duration flight activity after dark, although daytime activity is greater [100,110,185]. Isard et al. [58] interpreted cessation of western corn rootworm captures from 10-m towers at the approach of darkness as cessation of flight, but more properly, it meant ascent into the atmosphere during migration stopped. It is possible, even likely, that individuals ascending earlier in the evening maintained their flight at higher altitudes after dark during the transmigration phase, as is the case with a number of other migratory insect species [80,345]. In an interview, W.B. Showers described capturing western corn rootworm beetles on a television tower in light traps, which were not visible from the ground, suggesting migrants were flying at night [353]. Vörös et al. [354] reported captures of this species in light traps 2 km from the nearest corn.

#### 7.4.2. Transmigration Phase

Behavior of western corn rootworm during the transmigration phase is largely unknown. Many migrating insects ascend to a layer of maximum winds just below the temperature inversion during stable atmospheric conditions, which is often around 200–400 m a.g.l., but cruising altitudes can be much higher. Laboratory studies of western corn rootworm flight lack manipulation of atmospheric and wind conditions or any understanding of how a migrating beetle behaves once it enters such fast-moving parcels of air. The use of entomological radar to observe ascent and transmigration of this species may help fill many of our knowledge gaps, as it has for numerous other migratory insects [80,198,355,356,357]. To account for wash-ups of western corn rootworm beetles on Lake Michigan shores, Isard et al. [60] postulated that migrating beetles encountered the daytime lake breeze at 2 km in elevation before being mixed downward over the water into onshore winds. Grant and Seevers [59] cite unpublished data that beetles can be “found readily at heights of at least 35 m” near cornfields. Spencer et al. [148] cited a newspaper report of western corn rootworm beetles accumulating on tall buildings in Chicago at about 130 m a.g.l. Adults were captured in light traps on the television tower in central Iowa mentioned earlier at 76, 152, and 275 m a.g.l. [353]. Systematic sampling of this insect at intervals above 10 m would help clarify behavior during the transmigration phase, such as flight altitude in relation to atmospheric and wind conditions. It would also help elucidate the demographic makeup of the migrating population. For example, although a relatively small proportion of ascending beetles captured at 10 m are males, the flight durations of males making sustained flights on flight mills are less than those of females. We predict that the proportionate number of males declines with altitude. Sampling at high altitudes can be attempted via balloon netting [199,210,345] or netting using remote-controlled aircraft (e.g., [358,359]). Employment of entomological radar in conjunction with 10 m tower netting would be valuable in determining altitude of transmigratory flight by ascending western corn rootworm.

#### 7.4.3. Termination Phase

Little is known about the termination phase of western corn rootworm migratory flight. The lack of observational data for descent of migrating individuals leaves a troublesome hole in our knowledge. A cornfield out of which adults are captured at 10 m while ascending into the atmosphere represents a concentrated departure point, but spatial dilution of migrating beetle density inevitably occurs during the transmigration phase and during the termination phase itself as individuals descend after different durations of flight. On flight mills, sustained flights exhibit a fat-tailed distribution over a wide range of duration and distance [61,100,110,151,185], and vagaries in wind profiles at different flight altitudes and times will generate variation in descent locations as well. Because migratory flight is non-appetitive, differences in suitability of habitat will not serve to concentrate immigrants to particular cornfields in a landscape until after termination of migration and resumption of appetitive ranging flight. Whether this transition occurs before, during, or after descent is unknown, and undoubtedly affects the ultimate terminus of an immigrant.

#### 7.4.4. Number of Migratory Flights

Many migratory insects engage in more than one bout of migratory flight. As discussed earlier, it is unlikely that western corn rootworm migrants engage in more than one migratory flight in a single day because of the need for permissive atmospheric conditions which change as part of a diurnal cycle of insolation [58,68,167]. Migratory flights may occur on more than one day, a hypothesis deserving investigation. Such multi-leg or “stopover” migratory flights are common in long-distance, seasonal insect migrants between overwintering and reproductive ranges (e.g., [182,203,360]). However, it seems unlikely to occur more than once per lifetime in western corn rootworm. This is an aseasonal, partial migratory species where the evolutionary impetus behind migration presumably is displacement itself rather than a destination, and displacement can be achieved in a single bout of non-appetitive flight.

### 7.5. Additional Knowledge Gaps

There are a number of other major knowledge gaps that can and should be addressed experimentally. The genetic and physiological mechanisms underlying development of a western corn rootworm larva into a resident or migrant adult behavioral phenotype are unknown. Crowding increases average distances flown by young mated females on flight mills [100], but is this a reflection of migrant flight behavior, resident flight behavior, or both? Or does it reflect a facultative increase in the proportion of migrant phenotypes in the population? To what extent does the proportion of residents and migrants within a population and across a landscape differ between years and spatially within years? Does resistance to Bt Cry toxins or crop rotation affect flight behavior, including the proportion of residents and migrants in a population? Performance of resistant beetles in tethered flight compared to wild-type beetles could be valuable in understanding and predicting evolution, maintenance, and spread of resistance. What is the physiology of migrants during preparation for, during, and after migratory flight? Juvenile hormone (JH) is involved in expression of sustained flight in western corn rootworm [151], but little else is known regarding endocrine control of migration. Advances have been made in a few other insects in identifying the genetic underpinnings of insect flight and movement [92,325,361,362]. The most powerful approach combines tethered-flight assays with genomic and transcriptomic tools. While not trivial, tethered-flight experimentation with western corn rootworm is within reach of most laboratories [363]. The recent publication of this species’ transcriptome [364] and genome [365] will help make genetic studies of the biological components of western corn rootworm movement ecology far more tractable than they have been up to now.

## Figures and Tables

**Figure 1 insects-14-00922-f001:**
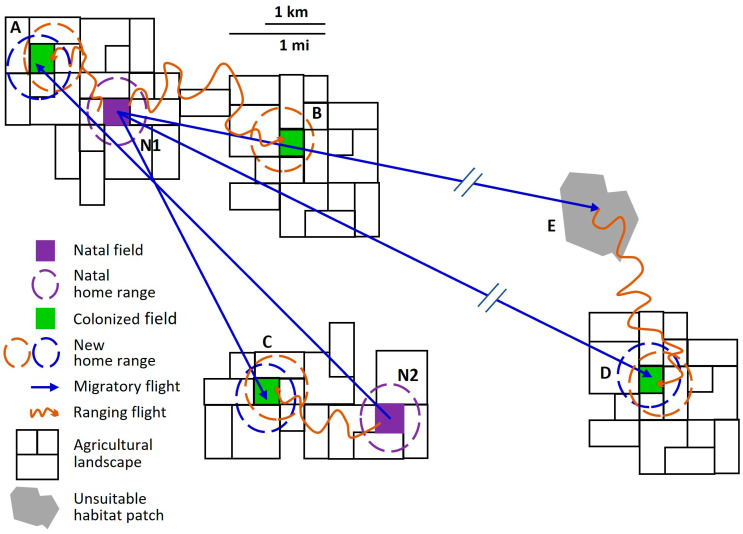
Schematic of types of flight behavior by *resident* and *migrant* phenotypes of western corn rootworm, and resulting spatial displacement across local and regional landscapes relative to natal and colonized cornfields. The schematic crop fields in the figure represent typical dimensions of rectangular fields in the U.S. Corn Belt, where the smallest squares shown are 0.25 mi (0.40 km) per side, or 40 acres (16.2 ha). In this depiction, western corn rootworm adults eclose in two natal fields (**N1** and **N2**). A *resident* may ultimately engage only in appetitive station-keeping behaviors throughout its lifetime, resulting in net displacement of only short distances within the natal field or after emigration into habitat in the natal field’s immediate vicinity. This area of station-keeping activity constitutes the natal home range (purple circle) and is observed as diffusive spread of the field’s population. Some residents engage in an appetitive ranging flight (meandering brown arrow) in search of a needed resource or better habitat, emigrating from the natal field and beyond the natal home range. Ranging flight is facultative and triggered by proximate conditions such as lack of a needed resource or deteriorating habitat. A ranging resident stops when it encounters the sought-after resource, which may be a relatively short distance away (e.g., **N1** → **A**, **N2** → **C**), or a longer distance across the local landscape (e.g., **N1** → **B**). In either case, after immigration into a suitable habitat, station-keeping behaviors resume in the colonized field, and a new home range (brown circle) emerges. A *migrant* adult emigrates from the natal field via a non-appetitive migratory flight (straight blue arrow) over relatively long distances, not only beyond the natal home range, but often beyond the local landscape (e.g., **N1** → **C**, **N1** → **D**, **N2** → **A**). Migration is not initiated in direct response to proximate conditions. Thus, migrants may emigrate from a highly suitable natal field in which a large number of residents remain to reproduce. Migration is innate to the migrant phenotype and is initiated in females during a narrow developmental window after mating but before egg maturation. Migratory flight is straight-line (non-meandering), and in western corn rootworm is not directed toward a geographic goal or in a preferred direction (e.g., beetles on opposite trajectories, such as **N1** → **D** and **N2** → **A**, can issue from the same field). The migrating insect does not respond to resource cues or cease flight when encountering suitable habitat (e.g., blue arrow passing over field **B**). Instead, migration is terminated in response to global environmental cues like sunset, or intrinsic cues such as an internal clock or physiological status, which have not yet been elucidated for western corn rootworm. If the insect fortuitously terminates its migratory flight in suitable habitat (e.g., fields **A**, **C**, and **D**), it then resumes station-keeping behavior, and a new home range emerges (blue circle). If migration terminates in unsuitable habitat (e.g., **N1** → **E**), the migrant initiates appetitive ranging behavior through the local landscape in search of appropriate habitat (e.g., **E** → **D**). Once suitable habitat is encountered, ranging ends, station-keeping behaviors resume, and a new home range emerges.

**Figure 2 insects-14-00922-f002:**
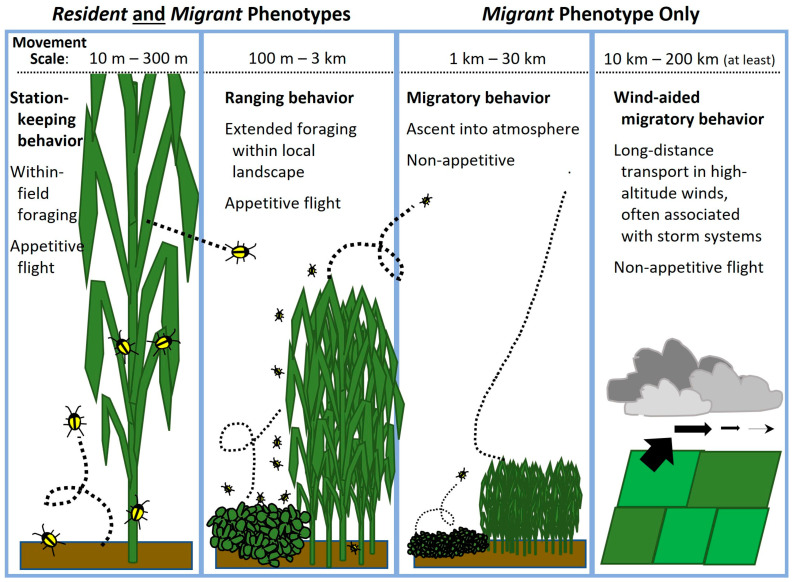
Approximate scales of net displacement during a bout of flight activity by western corn rootworm and assignment of behaviors to both *resident* and *migrant* phenotypes, or exclusively to the *migrant* phenotype. Movement behaviors [64] differ fundamentally as either appetitive or non-appetitive. Appetitive behaviors (station-keeping and ranging behaviors) are motivated by search for a needed resource. Non-appetitive behavior is migratory, motivated by the goal of spatial displacement itself. Migrant western corn rootworms begin migratory flight by purposely ascending high into the atmosphere where fast winds increase speed and distance of displacement.

**Table 1 insects-14-00922-t001:** Mated female western corn rootworm of indicated age ranges undertaking a sustained flight of ≥20 min in laboratory tethered-flight studies.

Study	Age (d)	No. Tested	No. Making Sustained Flight	% Making Sustained Flight
Coats et al. [61]	2–15	183	28	15
	2–9 ^a^	135	28	21
	2–7 ^a^	95	23	24
	5–6 ^a^	48	15	31
	10–15 ^a^	48	0	0
Coats et al. ^b^ [151]	2–11	97	28	29
	3–7 ^c^	54	28	52
	5–6 ^c^	28	15	54
	9–11 ^c^	27	0	0
Naranjo [110]	2–7	75 ^d^	18	24
	10–17	34	5 ^d^	15
	23–30	100 ^d^	4	4
Naranjo ^e^ [183]	5–10	77	19	25
	20–25	61	4	7
Wilson [184]	5–6	204	65 ^d^	32
Stebbing et al. [185]	3–20	54	15	28
Yu et al. ^f^ [100]	6	159	36	23

^a^ Subset of data included in the 2–15-d age group; ^b^ untreated mated female controls; ^c^ subset of data included in the 2–11-d age group; ^d^ inferred from reported data; ^e^ two sets of untreated controls for each age group, summed; ^f^ from data in Supplemental Material of Yu et al. [100].

## Data Availability

No new data were created or analyzed in this study. Data sharing is not applicable to this article.

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
