# Peer review of "Movement Ecology of Adult Western Corn Rootworm: Implications for Management"

_insects, 2023, doi:10.3390/insects14120922_

Round 1

Reviewer 1 Report

Comments and Suggestions for Authors

The review by Sappington and Spencer on the movement ecology of WCR adults was the most comprehensive review on any topic that I have ever read. While 356 citations may be a bit of an overkill, future generations of rootworm scientists will find themselves fortunate to find this information all in one place on this topic.  The writing is eloquent.  This review will be cited for decades and needs to be published. 

               I did double check every rootworm reference on whether it made sense to cite for the statement being documented.  A few errors were found:

Page 8, line 17 – the authors cited the wrong Ludwick et al. 2017.  They cited.

Ludwick, D.C.; Meihls, L.N.; Ostlie, K.R.; Potter, B.D.; French, L.; Hibbard, B.E. Minnesota field population of western corn 44 rootworm (Coleoptera: Chrysomelidae) shows incomplete resistance to Cry34Ab1/Cry35Ab1 and Cry3Bb1. J. Appl. Entomol. 45 2017, 141, 28–40. http://doi.org/10.1111/jen.12377.

However, for the statement being documented, they likely intended to cite

Ludwick, D.C., A. Zukoff, M. Higdon, and B.E. Hibbard.  2017.  Protandry of the western corn rootworm (Coleoptera: Chrysomelidae) partially due to earlier egg hatch of males J. Kans. Entomol. 90: 94-99.

Page 9 – lines 24-30. This point about WCR pheromone is important to the overall theme of the paper in that WCR were not in large monocultures until very recently in evolutionary time.  I like many things about this review and pointing this out is certainly one of them.

Page 21 – last paragraph. The cucurbitacin paragraph needs some rewriting because it is not accurate as written, especially in line 40 and 51 where mention of attraction and behavior are associated with cucurbitacins.  Cucurbitacins as a group are not volatile and therefore cannot be attractants.  It is the volatiles from Buffalo root powder that were attractive components from the bait. Interestingly, As noted in the Tallamy et al book chapter:

“LeConte (1868) based the species, D. virgifera, on two specimens found on ‘wild gourd’, so the association of WCR with members of the Cucurbitaceae dates back to the earliest published record for the species. Diabrotica spp., in general, have been associated with blossoms of varying Cucurbita spp. (Fronk and Slater, 1956; Howe and Rhodes, 1976; Bach, 1977; Fisher et al., 1984). Andersen and Metcalf (1987) examined selected Cucurbita spp. and cultivars for differences in floral volatile release, blossom cucurbitacin content and pollen content of male blossoms, and correlated these data to preference by Diabrotica beetles.” ….

“Although a behavioural bioassay is the only reliable means of evaluating an entire complement of compounds in a semiochemical blend, electroantennogram (EAG) recording is usually helpful in detecting the most important components in the blend. This technique is a simple method for electrophysiological detection of the responses of insect antennae to volatile semiochemicals (Roelofs, 1984). Hibbard et al. (1997b) used an EAG-driven isolation and identification schemes to identify (E,E)-3,5-octadien-2-one, (E,Z)-2,6-nonadienal, (E)-2-nonenal, 2-phenethanol, benzyl alcohol, and 6,10-dimethyl-5,9-undecadien-2-one from extracts of buffalo gourd (C. foetidissima Humboldt, Bonpland and Kunth) root powder, the behaviourally active component of semiochemical baits for adult corn rootworm control programmes. Cossé and Baker (1999) isolated several of these same compounds from buffalo gourd root powder and demonstrated that (E,E)-3,5-octadien-2-one was attractive to D. barberi adults.”

Page 22, line 43. Brackets [] are missing around the reference which should be “[29]”.

Page 31, line 22 – The Li et al. paper does not have a citation number here.  The number [46] needs to be added here.

Author Response

Reviewer 1:

Comment #1: I did double check every rootworm reference on whether it made sense to cite for the statement being documented.  A few errors were found:

Page 8, line 17 – the authors cited the wrong Ludwick et al. 2017.  They cited.

Ludwick, D.C.; Meihls, L.N.; Ostlie, K.R.; Potter, B.D.; French, L.; Hibbard, B.E. Minnesota field population of western corn 44 rootworm (Coleoptera: Chrysomelidae) shows incomplete resistance to Cry34Ab1/Cry35Ab1 and Cry3Bb1. J. Appl. Entomol. 45 2017, 141, 28–40. http://doi.org/10.1111/jen.12377.

However, for the statement being documented, they likely intended to cite

Ludwick, D.C., A. Zukoff, M. Higdon, and B.E. Hibbard.  2017.  Protandry of the western corn rootworm (Coleoptera: Chrysomelidae) partially due to earlier egg hatch of males J. Kans. Entomol. 90: 94-99.

Response: The reviewer is correct, and we corrected this reference on p.8, L52 and added the new citation [120] in the list of References.

Comment #2:  Page 21 – last paragraph. The cucurbitacin paragraph needs some rewriting because it is not accurate as written, especially in line 40 and 51 where mention of attraction and behavior are associated with cucurbitacins.  Cucurbitacins as a group are not volatile and therefore cannot be attractants.  It is the volatiles from Buffalo root powder that were attractive components from the bait.

Response: The reviewer is correct, we made a mistake, and we thank him/her for pointing this out! The word “attractant” has been changed to “arrestant” in original Line 40. After making this correction, original Line 51 is no longer incorrect, but to ensure clarity we have added a parenthetical comment. This part of the sentence (P22, L46-48) now reads, “…there was a significant decrease in behavioral response (as arrestant, feeding stimulant, or both) to cucurbitacin…”, still citing Siegfried et al. 2004 [269] who wrote about the decrease in behavioral response.

Comment #3:  Page 22, line 43. Brackets [] are missing around the reference which should be “[29]”.

 Response: Added brackets (now reference [36]), P.23, L40.

Comment #4:  Page 31, line 22 – The Li et al. paper does not have a citation number here.  The number [46] needs to be added here.

Response: Done. (now reference [251])

Reviewer 2 Report

Comments and Suggestions for Authors

Comments and suggestions are in paper.  The paper provides a comprehensive analysis of literary data presented in a qualitative manner, which enhances understanding of the topic.

Author Response

Reviewer 2:

Comments and suggestions are in paper.  The paper provides a comprehensive analysis of literary data presented in a qualitative manner, which enhances understanding of the topic.

Comment #1 [from note on manuscript, highlighting scale bar in Fig. 1]:  It is not clear the distance of 1 km from the figure 1. It should be better explained.

Response:  Added a 1-mile scale bar to the Figure, and the following sentence to the caption: “The schematic crop fields in the figure represent typical dimensions of rectangular fields in the U.S. Corn Belt, where the smallest squares shown are 0.25 mi (0.40 km) per side, or 40 acres (16.2 ha).”

Comment #2  [From note on manuscript, Fig. 2]:  Displacement scale is not clear. Could you write more clearly?

 Response:  We are not sure whether the reviewer found the terminology of “Displacement Scale” unclear or the scientific notation for the scale unclear, so we have changed both. We changed “Displacement Scale:” to “Movement Scale:”. And we simplified the values to indicate the ranges directly in m or km.

Reviewer 3 Report

Comments and Suggestions for Authors

The present manuscript contains a deep and complete review about the corn pest Diabrotica virgifera and the implications of the movement ecology for its management. Nowadays, in this situation of global change, generating knowledge about pest species related to its ecology, genetics, movement, resistances... becomes fundamental to succeed when control measures are applied.

Although the authors have carried out an extensive work, I have few comments about the manuscript submitted.

- In some places of the text you use the sentence "...its penchant to evolve resistance to many forms of control". I clearly understand the meaning of your sentence. However, to be more accurate, resistances are not developed because of the control measures, it sounds lamackist. Control measures perform a selection pressure that favours the survival (and thus, the reproduction) of those individuals with resistant genotypes/ phenotypes instead of those non resistant.

-Which are the main measure control used against this pest? Insecticides, just BT crops, culture rotation, IPM... You can find this information in some parts of the text, however it may be better to know this info from the beggining of the manuscript.

-You mention resistance alleles in many parts of the text. Which genes are related to this resistance? What is the frequency of resistant alleles in the populations? Does these resistances cause a fitness-cost to the individuals when the pressure is not applied?

- IRM and IPM acronyms are not explained.

-In point 2.Premating movement: To me sounds a little bit blurry the whole section. There are just 4 citations (115-117) but you mix them all the time and some times you can get lost.

- Point 3. Mating and movement, lines 46-50: I don't understand the genetic analysis performed in the reference cited and the relationship with survival. What did they measured?

-Point 4: Post mating movement, page 12 lines 23 to the end of the page. When you perform culture rotation and the population seems to be resistant. Do you know or have you ever measure the impact caused by the pest in the alternative  culture? The same/ lower/ higher?

-Point 5, section 5.3 line 23. you put 3.8 km/h but the international system is m/s. In this same section, lines 49-50 I don't see why you say that about gut content analysis.

- Range expansion: Do you find the same damage in both continents (Europe and America)? Given the restricted measure controls and the European Legislation about chemical products/ transgenic products (in case of BTcrops and the promotion of IPM), are there the same resistant strains and frequencies? Given that this pest is an introduction in Europe, does its presence affect other pest species in the field?  I mean, in Europe only Spains grows BT corn and the main pest described against this culture is Seoania nonagioides , do you know if there is any interaction among this two species?

-5.6 Spatial distribution of resistance. What are the genetic resistances? you mention strains and kind of resistances related to BT crops or crop rotation but there is not information about the genes or the alleles. In this same section, page 22 line 43 put the reference in brackets "St. Claire et al. [29] compared" instead of "St. Claire et al 29 compared..."

- You mentioned many locations across America and Europe. Have you ever considered to put a map with the cited places in order to facilitate the understanding of the text? Just to have a real reference about distances and places.

- All the genetic studies cited but one cited in table 2 have been performed with microsatellite markers. Did you find more recent studies with SNPs or mitochondrial markers?

-Figure 2: In Migrant phenotype only is correct the presence of the "+" in the displacement scale?.

-Conclusions and synthesis. You add more information to the review with more experiments. I think that 6 pages of conclusions and synthesis are too much. Consider reorder and put part of the information in the main text.

Author Response

Reviewer 3:

Comment #1 - In some places of the text you use the sentence "...its penchant to evolve resistance to many forms of control". I clearly understand the meaning of your sentence. However, to be more accurate, resistances are not developed because of the control measures, it sounds lamackist. Control measures perform a selection pressure that favours the survival (and thus, the reproduction) of those individuals with resistant genotypes/ phenotypes instead of those non resistant.

Response: We agree, although we did not expect this “literary” statement would be misconstrued as characterizing the process of natural selection in western corn rootworm. Rather it is a shorthand statement capturing a pattern of rapid resistance evolution that has repeated time after time when this species has been challenged by a management tactic. Nevertheless, we take the reviewer’s point, and have rephrased in the following places to be more accurate:

P.1, Lines 12-13: changed to, “It is difficult to manage, in part because practical field resistance has evolved to many forms of control.”

P.2, L20-21: changed to, “Nearly every management tactic deployed against western corn rootworm has been compromised by evolution of practical field resistance.”

P.23, L33-34: changed to,  “…the minimum necessary for resistance to evolve in a population…”

 Comment #2 -Which are the main measure control used against this pest? Insecticides, just BT crops, culture rotation, IPM... You can find this information in some parts of the text, however it may be better to know this info from the beggining of the manuscript.

Response:  We rearranged, added text, and divided this introductory material into two paragraphs. Because this information is covered in detail in other parts of the manuscript, we kept it short with a minimum of references, but should give the reader an initial idea of what control tactics are used. The added material P1, L14-19): “Annual crop rotation was recognized as an effective management tool since the earliest days of its emergence as a pest [Gillette 1912 [12]] and remains an important option today [Levine et al. 2002, Ludwick and Hibbard 2016, Carrière et al. 2020, Bažok et al. 2021, Furlan et al. 2022 [13-17]. In non-rotated corn in North America, larval control with soil insecticides was the most common management tool in the last half of the 20th century [Meinke et al. 2021 [18], until the advent of transgenic Bt-corn in 2003 [Gassmann et al. 2021 [19]]. Control of adults with foliar insecticides was a common tactic in parts of the Great Plains as well [Meinke et al. 2021 [18]].”

Comment #3 -You mention resistance alleles in many parts of the text. Which genes are related to this resistance? What is the frequency of resistant alleles in the populations? Does these resistances cause a fitness-cost to the individuals when the pressure is not applied?

Response:  In our treatment of resistance, as is standard practice in almost all insect resistance management literature, specific genes, alleles, or gene regions are neither invoked nor needed to describe and explore Bt resistance dynamics and management at the population-level. Assignment of resistance to particular gene sequences has been published for some types of resistance (e.g., Flagel et al. 2014), but identification of these putative genes has not led to more effective management. Fitness costs of resistance have also been examined; e.g., they are apparently weak in the case of Bt resistance (summarized in Gassmann et al. 2021). In the case of rotation resistance, it does clearly affect movement patterns (genes for rotation resistance are not identified - to our knowledge); however, our best understanding is more at the level of behavioral and physiological mechanisms. Rotation resistance is a complex condition that is unlikely governed by just one or a few genes. We know that there are gene expression differences (for protease inhibitors in rotation resistant WCR populations), differences in the suite of microbes present in the gut of resistant and susceptible populations, as well as behavioral differences upon exposure/consumption of host and non-host tissues. WCR behavioral adaption to crop rotation is likely polygenic.

All of that said, the details of specific Bt resistance alleles and fitness costs, and the research efforts to identify them, are beyond the scope of our review. They can affect spread of resistance, but our review is about rootworm behavior, not about resistance itself. Instead, we are interested in and describe certain observed patterns of resistance spread and spatial distributions for what they can tell us about rootworm movement behavior.

Our point in this regard is in the sentence at P.20, L14-19: “The same issues that can complicate inference of flight distances from the rate of the species range expansion also apply to rate of resistance expansion, but with additional complicating factors. The most important is that resistance is a phenotype of an individual [40] controlled by one or more genes, each with its own set of alleles, and the resistance phenotype is not a selectively neutral marker.”  Knowing the identity of resistance alleles would not change that assessment or our inferences about the rootworm’s movement ecology.

Comment #4 - IRM and IPM acronyms are not explained.

 Response:  (P. 8, L43) was the only place in the text where IPM was used, and is now replaced here by “pest management”. IRM is now defined at first mention (not including Simple Summary and Abstract) as the acronym for “insect resistance management”, P.8, L4.

 Comment #5 - Point 2. Premating movement: To me sounds a little bit blurry the whole section. There are just 4 citations (115-117) but you mix them all the time and sometimes you can get lost.

 Response:  Thank you for the comment and your concern about the “blurriness” of Section 2.

Prior to the development of Bt corn targeting corn rootworm beetles, there were no detailed field scale evaluations of western corn rootworm movement and mating. Uncertainty about the behavior of mate-seeking male and female western corn rootworms in Bt and non-Bt (refuge) corn stimulated intra- and interfield movement research. The Spencer et al. [22] study was one of the first to use a marking technique that could measure western corn rootworm intrafield movement at the field scale. Subsequent studies (Spencer et al. [115] and Hughson [116]) applied the Cry protein detection method to track western corn rootworm behavior (mating and movement) in fields with actual refuges and Bt corn. They were the first evaluations of ‘refuge” function designed to observe how beetles behaved in real refuges and followed them season-long. Work by Taylor and Krupke [117] used 15-Nitrogen isotope to definitively label insects that developed on Bt or non-Bt (refuge) plants. Other than an earlier Taylor et al. (2017) study that looked at mating probabilities between beetles that developed on Bt and non-Bt plants in small enclosures over 8-row plots (unsuitable for movement studies), the cited works [22, 115, 116, 117] represent the extent of mating and movement quantification at a scale relevant to management.

To address the blurriness, we have extensively revised Section 2 (P8, L49 through P10, L6) with the aim to improve the narrative quality, consolidate similar content, and provide clearer context.

 Comment #6 - Point 3. Mating and movement, lines 46-50: I don't understand the genetic analysis performed in the reference cited and the relationship with survival. What did they measured?

 Response: These two sentences were needlessly complicated by our mention of the authors’ [134] (now reference [147]) study of admixed invasive populations for fitness differences. The main point of one of their findings that is actually relevant to our discussion in the text is now summarized in a single sentence (P10, L3841): “Bermond et al. [147] found 3-fold greater survival of females than males under starvation in the laboratory, and hypothesized that female use of nutrients from the male spermatophore may explain this result.”

 Comment #7 - Point 4: Post mating movement, page 12 lines 23 to the end of the page. When you perform culture rotation and the population seems to be resistant. Do you know or have you ever measure the impact caused by the pest in the alternative culture? The same/ lower/ higher?

 Response: Though western corn rootworm egglaying in soybean can cause devastating yield losses in rotated corn, the larvae cannot survive on soybean roots and have no impact on that crop. The impact of adult herbivory on soybean yield is likely low and confounded with that of other resident herbivores. While rotation-resistant western corn rootworm can become abundant and feed on foliage while in the rotated crop (almost exclusively soybean in the U.S. Corn Belt) [13], the aggregate defoliation risk from other resident soybean pests (e.g. Japanese beetles, green cloverworms, bean leaf beetles, grasshoppers, and etc.) is likely greater. Furthermore, soybean plants can tolerate significant defoliation (20%-30% - depending on their phenology) before suffering yield loss [ https://crops.extension.iastate.edu/cropnews/2023/07/keep-eye-out-soybean-defoliators] and it would be unusual for herbivory by any single pest to cause economic damage (defoliation-caused economic damage). After the discovery of rotation-resistant western corn rootworm, adult control (with broadcast insecticide) was suggested as a rootworm management tactic, not to protect soybean yield, but rather to prevent WCR oviposition in soybean. This failed due to frequent interfield movement of rotation-resistant WCR which allowed rootworm populations to rebound within 1-2 weeks of an application. An evaluation of early or late season pyrethroid insecticide application in soybean to kill egglaying WCR adults (and thus reduce the potential for larval injury in rotated corn) found no significant impact of sprays on subsequent injury in rotated corn (Kaluf, A.L. 2016. “Evaluating the effect of foliar insecticides on suspected Bt-resistant western corn rootworm in rotated soybean”, M.S. Thesis, Dept. of Crop Sciences, University of Illinois, Urbana-Champaign http://hdl.handle.net/2142/90663)

We have added a sentence to the text at the end of the next paragraph (Page 14, L12-14) to address this: “While rotation-resistant western corn rootworm herbivory in soybean can sometimes be dramatic [13], the high soybean tolerance for defoliation [180] suggests rootworm activity is unlikely to affect soybean yield, without contributions by other resident soybean herbivores.“

Comment #8 - Point 5:  section 5.3 line 23. you put 3.8 km/h but the international system ism/s.

 Response: Given that we have presented distance measurements for this topic in km, sharing the flight mill rate in km/h seems most clear and we would like to retain it. However, we have added “(1.06 m/s)” in parentheses after “3.8 km/h”. P17, L17.

Comment #9 - Point 5:  In this same section, lines 49-50 I don't see why you say that about gut content analysis.

 Response: Mention of gut contents is significant because we were able to identify the presence of Bt proteins (serving as a marker) from upwind locations – further evidence for local ascent. We have connected the two sentences to make this clear (P17, L42-44): “The females flying at 10 m were also likely of local origin as analysis of their gut contents detected the presence of Bt proteins (or other plant tissues) available in upwind source fields [58,211].”

Comment #10 - Range expansion: Do you find the same damage in both continents (Europe and America)? Given the restricted measure controls and the European Legislation about chemical products/ transgenic products (in case of BT crops and the promotion of IPM), are there the same resistant strains and frequencies? Given that this pest is an introduction in Europe, does its presence affect other pest species in the field? I mean, in Europe only Spains grows BT corn and the main pest described against this culture is Seoania nonagioides , do you know if there is any interaction among this two species?

 Response: This is a level of detail beyond the scope of our review. We are simply looking for any evidence, direct or indirect, that provides clues about western corn rootworm movement behavior and patterns. We suggest consulting Bažok et al. 2021 [16] and papers cited therein for up-to-date details about WCR management in the EU.

Comment #11 – Section 5.6 Spatial distribution of resistance. What are the genetic resistances? Yo mention strains and kind of resistances related to BT crops or crop rotation but there is not information about the genes or the alleles. In this same section, page 22 line 43 put the reference in brackets "St. Claire et al. [29] compared" instead of "St. Claire et al 29 compared..."

Response:  Please see the response to Comment #3 above regarding the issue of identification of resistance alleles.

Added brackets (now reference [36]), P.23, L40.

 Comment #12 - You mentioned many locations across America and Europe. Have you ever considered to put a map with the cited places in order to facilitate the understanding of the text? Just to have a real reference about distances and places.

 Response: Yes, we have thought about this and agree that it could add some value. However, we are very hesitant to try this. Given that this is already an exceptionally long paper, and considering the amount of time it would take to prepare such a map large enough and detailed enough to resolve locations across so many regions and studies, the idea seems not feasible, and not strictly necessary.

 Comment #13 - All the genetic studies cited but one cited in table 2 have been performed with microsatellite markers. Did you find more recent studies with SNPs or mitochondrial markers?

 Response: To our knowledge, no more recent population genetics studies have been published on this species. We and coauthors have completed a population genetics study of western corn rootworm using large numbers of genomic SNPs which is near submission, but cannot be cited yet; it is alluded to in the manuscript as “(T.W.S. and J.L.S, unpublished data)” on P.28, L7. There is also a paper by Coates et al. 2009 [313] that used SNPs, mentioned in the Citation column of Table 2 for the first entry, but is not listed separately because it examined mostly the same populations as Kim and Sappington [306]. The purpose of the Coates et al. [313] paper was to compare results between microsatellites and SNPs on the same dataset; the results were similar between the markers, so we do not think it warrants a separate entry. Note also that AFLPs were used in addition to microsatellites by Miller et al. [314], as listed in the second entry in the table.

 -Figure 2: In Migrant phenotype only is correct the presence of the "+" in the displacement scale?

 Response: The “+” was meant to convey the fact that 3 x 10^5 m (300 km) is not a strict upper bound of western corn rootworm movement capacity, but that it may be even greater. We have made a number of changes to the headers in the figure including substituting values in m and km instead of scientific notation. The 300 km value was a mistake, it should have been 200 km, supported by range expansion data; although it is likely that movement is at least 300 km based on the argument on P.18, L44-49, and as yet unpublished population genetics data, there is published data supporting 200 km so we have changed that value. However, because the range expansion data is surely an underestimate as described in the text, we have replaced the entry in the final column of the figure with “10 km – 200 km (at least)”, which should be much clearer as to meaning.

 Comment #14 - Conclusions and synthesis. You add more information to the review with more experiments. I think that 6 pages of conclusions and synthesis are too much. Consider reorder and put part of the information in the main text.

Response: The “Conclusions and Synthesis” (Section 6) is not in itself unusually long, but we agree that in combination with Section 7 (Unanswered questions and future directions) it is longer than most readers expect. The details in these two sections are more than a list of vague ideas for future experiments. The background behind the unanswered questions is thoroughly discussed, which we feel is a strength. That said, we have made some changes: To help the reader navigate the material we have added multiple subheadings and subsubheadings to Section 7 delineating the issues being discussed. In addition, material related to more general information was moved to sections earlier in the paper, where they fit as well or better than at the end. These include two paragraphs related to partial migration possibly serving as a bet-hedging strategy (now P.7, L1-28), and introducing the three phases of migratory flight in the atmosphere (now P.5, L16-21).